# SHARP: Blockchain-Powered WSNs for Real-Time Student Health Monitoring and Personalized Learning

**DOI:** 10.3390/s25164885

**Published:** 2025-08-08

**Authors:** Zeqiang Xie, Zijian Li, Xinbing Liu

**Affiliations:** 1College of Artificial Intelligence, Dalian Maritime University, Dalian 116026, China; zeqiang@dlmu.edu.cn; 2Faculty of Artificial Intelligence, Universiti Teknologi Malaysia, Kuala Lumpur 54100, Malaysia; xinbing@graduate.utm.my

**Keywords:** smart education, student health monitoring, wearable sensors, wireless sensor networks, reinforcement learning, proof-of-authority blockchain

## Abstract

With the rapid advancement of the Internet of Things (IoT), artificial intelligence (AI), and blockchain technologies, educational research has increasingly explored smart and personalized learning systems. However, current approaches often suffer from fragmented integration of health monitoring and instructional adaptation, insufficient prediction accuracy of physiological states, and unresolved concerns regarding data privacy and security. To address these challenges, this study introduces SHARP, a novel blockchain-enhanced wireless sensor networks (WSNs) framework designed for real-time student health monitoring and personalized learning in smart educational environments. Wearable sensors enable continuous collection of physiological data, including heart rate variability, body temperature, and stress indicators. A deep neural network (DNN) processes these inputs to detect students’ physical and affective states, while a reinforcement learning (RL) algorithm dynamically generates individualised educational recommendations. A Proof-of-Authority (PoA) blockchain ensures secure, immutable, and transparent data management. Preliminary evaluations in simulated smart classrooms demonstrate significant improvements: the DNN achieves a 94.2% F1-score in state recognition, the RL module reduces critical event response latency, and energy efficiency improves by 23.5% compared to conventional baselines. Notably, intervention groups exhibit a 156% improvement in quiz scores over control groups. Compared to existing solutions, SHARP uniquely integrates multi-sensor physiological monitoring, real-time AI-based personalization, and blockchain-secured data governance in a unified framework. This results in superior accuracy, higher energy efficiency, and enhanced data integrity compared to prior IoT-based educational platforms. By combining intelligent sensing, adaptive analytics, and secure storage, SHARP offers a scalable and privacy-preserving solution for next-generation smart education.

## 1. Introduction

The rapid evolution of educational technologies has highlighted the growing importance of intelligent systems in facilitating personalized learning and promoting student well-being [1]. Conventional pedagogical approaches frequently fail to account for the heterogeneous physiological and cognitive states of learners, despite their demonstrated influence on knowledge acquisition and educational outcomes [2]. The development of real-time student monitoring systems with proactive intervention capabilities has emerged as a crucial requirement in modern education [3]. However, current implementations encounter substantial limitations, including fragmented integration of health monitoring and adaptive learning systems [4], unresolved data privacy issues, scalability constraints, and suboptimal accuracy in health-state assessment and pedagogical decision-making [5,6].

Currently, health monitoring solutions primarily rely on manual assessments or simplistic monitoring devices [7,8], providing only superficial data without comprehensive insights. Additionally, traditional approaches lack robust analytical tools and methods necessary for accurately predicting potential health issues or recommending personalized educational interventions [9]. Long-term use of wearable sensors in classrooms may raise concerns about physical discomfort or potential health effects, especially for younger students. Past incidents, such as the 2019 “surveillance headband” case in Jinhua, China, have sparked public debate on student privacy and psychological stress. These issues highlight the importance of ethical considerations and stakeholder acceptance in educational technology design. Centralised data management systems utilised in existing technologies pose further risks related to data security and user privacy, potentially exposing sensitive personal information to unauthorised access and misuse [10].

Real-time health monitoring and personalized learning are not just desirable features—they are increasingly essential in post-pandemic smart education systems. Mental fatigue, stress, and disengagement among students have been shown to correlate with poor academic outcomes and long-term health issues. However, traditional systems lack the sensing granularity and real-time feedback capabilities to support fine-grained interventions. Moreover, privacy concerns and system scalability further complicate large-scale deployments in educational institutions.

Emerging technologies, including Internet of Things (IoT), AI, and blockchain, offer promising pathways to address these limitations. However, the integration of these technologies in educational contexts remains underexplored, particularly regarding real-time health monitoring coupled with adaptive learning systems [11]. Existing studies on IoT-based monitoring often focus on single parameters like attendance or basic physical activities [12], failing to incorporate comprehensive physiological health monitoring integrated with learning processes. AI applications in education predominantly focus on academic performance prediction and rarely combine real-time physiological data analysis with adaptive instructional methods [13]. Moreover, while blockchain technology has been extensively studied for secure transactions and data integrity in various sectors, its potential in securing sensitive student health and educational data has not been thoroughly explored within education environments [14].

This research aims to bridge these gaps by developing an integrated, secure, and intelligent student health monitoring framework explicitly tailored for personalized educational contexts. The proposed framework harnesses advanced WSNs for continuous [15], real-time health data acquisition, coupled with robust AI-driven analytics and blockchain technology for secure, transparent, and immutable data handling [16,17].

This research introduces an unprecedented integration of wireless sensor networks, AI-based analytics, and blockchain technology into a unified [18,19], intelligent student health monitoring and personalized educational system. The developed platform addresses critical real-time health monitoring needs within educational environments, enhances privacy and security of sensitive data through the blockchain, and substantially improves the personalization and accuracy of educational interventions using advanced AI algorithms.

While technologies such as physiological monitoring and AI-based analysis are widely adopted in the medical field, their application in education requires heightened ethical scrutiny. This is due to the inherent power imbalance, the absence of strict regulatory frameworks, and the potential impact on students’ autonomy and psychological well-being. However, the SHARP framework is designed with voluntary participation, informed consent, and non-invasive data handling. Unlike clinical surveillance, our educational deployment emphasises empowerment, personalized support, and student control over their data. These principles help align the framework with ethical standards even in the absence of specific legislation.

In summary, the main contributions of our work are as follows:A ZigBee-based wireless sensor network framework designed to achieve continuous, real-time collection and efficient transmission of comprehensive physiological data within educational environments.Advanced AI algorithms, including machine learning models and neural networks, designed to accurately detect, analyse, and predict students’ physiological and psychological states, enabling proactive educational interventions and health monitoring.The implementation and validation of blockchain technology for secure, transparent, and immutable management of sensitive student health and academic data, significantly enhancing data privacy, security, and trust among stakeholders.

Overall, the proposed framework demonstrates substantial practical value by offering a holistic solution to enhance student health monitoring and educational personalization, thereby significantly contributing to the advancement of smart education paradigms.

Despite the growing interest in intelligent educational environments, two core challenges remain largely unaddressed: (1) the lack of reliable and secure real-time health monitoring infrastructures within classrooms, and (2) the absence of dynamic, personalized learning systems that adapt to students’ physiological and cognitive states. Existing solutions often rely on delayed cloud-based processing, static learning profiles, or lack privacy-preserving mechanisms. These limitations hinder timely intervention for at-risk students and fail to support adaptive learning at scale. Addressing these challenges is crucial for ensuring student well-being, improving engagement, and enabling data-driven pedagogical strategies.

To address the above challenges, this paper proposes SHARP, a blockchain-enhanced wireless sensor network framework for real-time student health monitoring and personalized learning. As illustrated in Figure 1, SHARP comprises a three-layer architecture that integrates physiological sensing, AI-driven personalization, and secure data management.

The remainder of this paper is organised as follows. Section 2 surveys the related work. Section 3 introduces the proposed methodology and theoretical model, detailing the framework architecture, ZigBee-based WSNs’ optimisation, AI-driven health-state prediction, blockchain-secured data management, reinforcement-learning-based personalization, and the cross-layer optimisation workflow. Section 4 describes the experimental design and evaluation, including the smart-classroom testbed, comparative baselines, and an extensive results analysis covering sensor ablation, prediction accuracy, and WSNs’ performance, scalability, security, and educational impact. Finally, Section 5 concludes this paper and outlines directions for future work.

## 2. Related Work

The integration of the IoT in educational environments has led to promising developments in smart learning spaces. For instance, Zhang et al. [20] proposed a sensor-driven framework that leverages ambient intelligence to enhance classroom interactivity. Similarly, Tabuenca et al. [21] and García-Monge et al. [22] focused on environmental monitoring using IoT, demonstrating improvements in energy efficiency and contextual awareness. Hanum et al. [23] extended this work by highlighting IoT’s role in resource and infrastructure management across educational institutions.

However, these prior efforts primarily concentrate on static environmental or administrative parameters and largely overlook the dynamic physiological states of individual learners—such as stress, fatigue, or engagement—which are critical to personalized education. Moreover, they do not address how such physiological data can be intelligently analysed and securely stored to enable adaptive, privacy-preserving learning systems. Blockchain ensures transparency and data integrity through tamper-resistant ledgers. To preserve privacy, our system stores only anonymised metadata on-chain while keeping identifiable physiological data encrypted off-chain.

Our work builds upon these foundations by incorporating real-time physiological monitoring, deep neural network (DNN)-based state prediction, and blockchain-enabled data integrity into a unified, student-centred architecture.

Physiological sensing has increasingly been recognised as a valuable tool to enhance educational experiences by providing insight into students’ cognitive and emotional states. Liu et al. [24] conducted a systematic review of wearable devices in education, highlighting the use of electrocardiogram (ECG) and electrodermal activity (EDA) sensors for estimating cognitive load. Anders et al. [25] investigated heart rate variability and academic performance, though their approach focused on intermittent measurements in controlled settings. These studies underscore the potential of physiological data but fall short in achieving continuous, real-time monitoring and integration with adaptive learning systems.

Recent IoT-based frameworks have demonstrated the feasibility of real-time physiological monitoring, particularly in healthcare. Awotunde et al. [26] proposed a cloud-IoT platform for smart healthcare monitoring, while Secara et al. [27] developed personalized health systems integrating wearable sensors and AI. However, these solutions are primarily tailored for clinical settings and lack the pedagogical and psychological considerations required in educational environments, such as unobtrusiveness and alignment with personalized learning interventions.

Moreover, the secure handling of sensitive physiological data remains an underexplored challenge. While blockchain technology has been applied for immutable health data logging, most solutions remain isolated from intelligent analytics and feedback mechanisms. To our knowledge, no existing system holistically integrates real-time sensing, personalized learning analytics, and decentralised data protection in smart classroom environments. Our proposed SHARP framework fills this gap by combining multi-modal wearable sensing, AI-based recommendation engines, and blockchain-secured data pipelines into a unified, responsive educational platform.

Artificial intelligence (AI) has gained significant traction in the education domain, especially for analysing student behaviour and enhancing engagement. Goldberg et al. [28] and Alruwais et al. [29] utilised machine learning to detect behavioural indicators such as attention and participation in real time. Building on this, Sayed et al. [30] and Feng et al. [31] proposed adaptive systems that tailor content delivery based on students’ cognitive states, leveraging neural networks and multimodal analytics. These studies demonstrate the potential of AI to personalize educational pathways by interpreting observable behavioural signals.

In parallel, predictive analytics and data mining techniques have been applied to forecast academic outcomes and support decision-making in education. Batool et al. [32] and Sarker et al. [33] reviewed data-driven models for academic performance prediction, while Ayeni et al. [34] and Yekollu et al. [35] emphasised personalized learning trajectories using AI-powered platforms. However, these systems predominantly rely on academic records and behavioural observations, often neglecting physiological and psychological data that can offer deeper insights into students’ real-time learning readiness. Our work addresses this gap by integrating AI-based prediction models with physiological sensing to enable more holistic, health-aware personalized learning in smart educational environments.

Blockchain technology has increasingly been explored to enhance security and transparency in educational data management. Balobaid et al. [36] proposed a blockchain-based model to ensure the immutability of student records, while Alsobhi et al. [37] and Ayub Khan et al. [38] demonstrated the potential of decentralized systems for credential verification and cross-institutional sharing. These efforts highlight the value of blockchain in preserving data ownership and verifiability. In this context, “sensitive data” refers to fine-grained physiological signals collected from students (e.g., heart rate variability, stress indices, and fatigue levels), which may reveal private emotional or medical conditions when exposed.

Beyond education, blockchain has also shown promise in broader IoT and healthcare contexts. Hayat et al. [39] applied blockchain to mitigate DDoS attacks in IoT networks, while Yaqoob et al. [40] demonstrated how blockchain can secure sensitive medical data, improving both privacy and auditability.

However, existing blockchain applications largely treat educational and physiological domains separately. To our knowledge, no prior work has integrated real-time physiological monitoring with educational blockchain systems. Our SHARP framework addresses this gap by employing blockchain to securely log and verify students’ health indicators and personalized learning recommendations, achieving both traceability and privacy preservation in a unified platform.

Comprehensive smart education systems that integrate multiple technologies remain relatively underexplored. Notable exceptions include the work of Benomar et al. [41], who developed a cloud-based architecture for connecting IoT devices in educational settings, and Leite et al. [42], who proposed theoretical frameworks for orchestrating technology-enhanced learning environments. Our research advances this domain by implementing a fully integrated framework that coordinates physiological monitoring, AI-driven analysis, and secure data management within a unified platform.

Recent studies have increasingly emphasised the integration of intelligent systems in education, particularly those leveraging multimodal data and privacy-aware architectures. Ahmad et al. [13] provided a comprehensive review on data-driven AI in education, highlighting the need for integrating affective computing and adaptive feedback systems to improve learning outcomes. Rehan [14] proposed a secure cloud–AI architecture for personalized learning, but their work lacked real-time physiological monitoring or decentralised data validation. Feng et al. [31] introduced a fuzzy neural network to assess teaching quality using multimodal inputs, yet it focused primarily on cognitive indicators without real-time health analytics. Alsobhi et al. [37] applied blockchain to academic micro-credentialing, demonstrating traceability but not addressing continuous biometric privacy. Kumar et al. [15] explored trust-aware blockchain integration in IoT healthcare, offering relevant insights into secure physiological data handling, though their context was limited to clinical settings.

Compared to these approaches, our SHARP framework uniquely integrates real-time multi-sensor health monitoring, adaptive reinforcement learning, and Proof-of-Authority blockchain consensus into a unified educational system. This positions our work at the intersection of smart sensing, ethical personalization, and secure learning environments, bridging the gap between affective computing, student empowerment, and data accountability in modern education.

Therefore, existing studies reveal clear gaps: limited integration between comprehensive physiological health monitoring and personalized educational interventions, insufficient real-time predictive analytics, and inadequate secure management of sensitive student data within educational environments. Our proposed research addresses these limitations by developing an integrated framework leveraging IoT for comprehensive health monitoring, AI for predictive and adaptive learning interventions, and blockchain for secure, transparent data handling, significantly advancing current educational technology capabilities.

## 3. Methodology and Theoretical Model

This section describes the proposed methodology, including framework architecture, AI-based analysis, and blockchain integration. A comprehensive mathematical model that underpins the AI-driven personalized interventions is presented. The main symbols used throughout this paper are summarised in Table 1.

The proposed integration of IoT, AI, and blockchain is not arbitrary but addresses distinct and interrelated challenges in real-time student health monitoring and adaptive learning. IoT wearable sensors enable continuous, non-intrusive collection of physiological signals such as heart rate and temperature, which are essential for capturing dynamic changes in students’ physical and cognitive states. AI techniques, profound neural networks, and reinforcement learning are employed to process these high-dimensional signals and generate personalized educational feedback that evolves. Blockchain, in turn, provides a secure, tamper-proof infrastructure for storing sensitive health data and recommendation records, ensuring data transparency, auditability, and trustworthiness in multi-stakeholder educational settings. Together, these technologies form a cohesive architecture where sensing, reasoning, and secure decision recording reinforce one another to support a privacy-aware and responsive smart education system.

Real-time health monitoring requires frequent and continuous transmission of high-resolution physiological data, which places a substantial burden on energy-constrained wireless sensor nodes. To ensure system longevity and stability, especially in large-scale deployments such as smart campuses, energy-efficient data transmission becomes a critical requirement. Therefore, we introduce a network optimisation model to minimise total energy consumption while maintaining data fidelity and low latency. This optimisation directly supports the system’s ability to perform uninterrupted real-time monitoring over extended periods.

Although numerous studies have addressed energy-efficient routing in WSNs, most focus on static sensor applications such as environmental monitoring or surveillance. These scenarios typically involve low-frequency, periodic data reporting and do not require stringent real-time performance.

In contrast, the SHARP involves continuous and high-frequency physiological data transmission from wearable devices. This unique context introduces new constraints, such as latency sensitivity, data burstiness, and varying sensor activation depending on students’ physiological changes.

Therefore, we design a tailored optimisation model that jointly considers energy usage, bandwidth availability, and latency constraints to dynamically adapt to changing educational and health-related data requirements.

### 3.1. Framework Architecture

The framework adopts a three-layer architecture as shown in Figure 1, namely (i) the Perception Network Layer, responsible for data collection via wearable sensors; (ii) the Analytics Control Layer, where AI models analyse physiological signals; and (iii) the Application Layer, which ensures secure data handling through blockchain mechanisms and provides personalized educational feedback.

The framework architecture consists of three core layers: Perception Network Layer, Analytics Control Layer, and Application Layer. Wearable sensors collect students’ physiological data continuously and transmit the data via WSNs based on ZigBee technology and IEEE802.15.4 protocols. The Perception Layer employs sensors such as photoplethysmography (PPG), temperature sensors, and electrodermal activity (EDA) sensors for comprehensive physiological data collection.

In contrast to superficial behavioural indicators, physiological signals such as heart rate variability, skin temperature, and stress-related biomarkers offer richer insight into students’ real-time cognitive load and emotional states. These bio-signals, when continuously monitored, enable proactive intervention before learning degradation becomes observable.

Prediction of student stress or fatigue enables proactive educational adaptation—before performance deteriorates. By forecasting likely short-term cognitive states, the system can trigger preventive strategies, thus maintaining engagement and avoiding negative learning spirals.

The proposed framework constitutes a holistic solution by tightly integrating three interdependent layers:

Perception Network Layer (IoT Sensing): This layer captures real-time physiological and behavioural data through wearable WSNs nodes.

Analytics Control Layer (AI Processing): DNNs interpret incoming signals to detect latent physical and emotional states, while reinforcement learning (RL) continuously adapts learning recommendations based on predicted states and feedback.

Application Layer (Blockchain Management): Blockchain ensures secure, transparent, and tamper-resistant storage of sensitive data and AI-driven interventions, fostering trust among stakeholders.

These layers interact seamlessly: data flows from sensors to AI models for real-time prediction and is securely recorded on-chain for traceability and policy enforcement. This vertical integration across sensing, decision-making, and trustworthy execution forms the basis for calling SHARP a holistic and problem-driven architecture.

### 3.2. Perception Network Layer: WSNs’ Sensing Architecture

The wireless sensor network operates using the IEEE802.15.4 standard [43] and ZigBee protocols to form a robust mesh network. Mathematically, the WSNs [44] are represented as a graph G=(V,E), where *V* is the set of sensor nodes, and *E* is the set of communication links. Each link $e_{ij}\in E$ between node *i* and *j* has associated attributes such as energy consumption $E_{ij}$, bandwidth $B_{ij}$, and latency $L_{ij}$. The network optimisation model seeks to minimise total energy consumption while ensuring data transmission quality, formulated as follows:(1)$$\begin{array}{c} \min\sum_{(i,j)\in E}E_{ij}x_{ij} \end{array}$$
subject to:(2)$$\begin{array}{c} \sum_{j:(i,j)\in E}x_{ij}-\sum_{j:(j,i)\in E}x_{ji}=\left\{\begin{array}{c} 1\\ -1\\ 0 \end{array}\right. \end{array}$$

If node i=1 it is a source node, if node i=−1 it is a sink node, and i=0 otherwise. $x_{ij}\in \left\{0,1\right\}$, ∀(i,j)∈E, where $x_{ij}$ indicates whether link $e_{ij}$ is selected for data transmission.

**Assumption 1.** 
*All sensor nodes are energy-constrained, and each transmission path must guarantee minimal energy usage without exceeding predefined latency limits.*

*The proposed WSN adopts an adaptive multi-hop routing protocol to optimise energy usage, reduce transmission delay, and maintain network reliability. Each path is evaluated using a composite cost function defined as follows:*

(3)
$$C_{ij}=\alpha E_{ij}+\beta D_{ij}+\gamma \left(1-R_{ij}\right)$$


*where*

*$E_{ij}$ denotes the normalised energy consumption for data transmission from node i to node j;*

*$D_{ij}$ represents the expected end-to-end delay;*

*$R_{ij}$ indicates the link reliability (packet delivery ratio);*

*α,β,γ are weighting coefficients satisfying α+β+γ=1 and adjusted based on task priority.*



To avoid overloading specific nodes with frequent transmissions, a rotation-based route scheduling mechanism is adopted. This ensures load balancing and prevents premature node failure by periodically alternating among the top-ranked paths based on a dynamic utility threshold.

Although the classroom scenario is spatially limited, direct transmission to the sink node is often suboptimal due to dynamic interference (e.g., human obstruction; multipath fading) and non-uniform node placement. Hence, a controlled multi-hop strategy offers better reliability and energy efficiency under practical deployment conditions.

### 3.3. Analytics Control Layer: AI-Based Signal Processing and Feedback

To precisely analyse and predict students’ health conditions, we employ advanced AI methodologies, specifically DNNs. We selected DNNs due to their strong capability in modelling complex, nonlinear relationships between multimodal physiological signals (e.g., ECG, EDA, HR, and BT) and latent psychological or physical states. Unlike shallow classifiers (e.g., SVM; decision trees), DNNs can automatically extract high-level representations from raw signals, improving generalisation for diverse student populations and varied learning contexts.

Prior work has demonstrated that DNNs outperform traditional models in detecting mental fatigue, stress, and abnormal heart rate variability in real-time biosignal analysis [45,46]. Identifying such states enables adaptive learning interventions, such as real-time workload adjustment or attention prompts. As these indicators may exhibit nonlinear and overlapping signal features, DNNs are particularly suited to capture these subtle patterns. The DNN model consists of an input layer, multiple hidden layers, and an output layer. The input features *X* represent physiological parameters, such as heart rate (HR), body temperature (BT), stress levels (SL), and fatigue indices (FI):(4)$$\begin{array}{c} X={\left[HR,BT,SL,FI\right]}^{T} \end{array}$$

The hidden layers employ activation functions (Rectified Linear Unit—ReLU) to learn nonlinear patterns:(5)Hi=ReLU(WiHi−1+bi),i=1,2,…L
where $H_{0}=X$, and $W_{i}$ and $b_{i}$ are weights and biases, respectively, and *L* represents the number of hidden layers. The output layer employs a softmax activation function to classify student health states into predefined categories (Normal, Attention Required, and Critical):(6)$$\begin{array}{c} Y=soft\max(W_{L+1}H_{L}+b_{L+1}) \end{array}$$

The objective function is defined by the cross-entropy loss to optimise prediction accuracy:(7)$$\begin{array}{c} J\left(\theta \right)=-\frac{1}{N}\sum_{i=1}^{N}\sum_{k=1}^{K}y_{ik}\log\left({\hat{y}}_{ik}\right) \end{array}$$
where *N* is the number of samples, *K* is the number of categories, $y_{ik}$ is the actual label, and ${\hat{y}}_{ik}$ is the predicted probability.

**Lemma 1.** 
*Given sufficient training data and proper regularization, the proposed DNN model converges to a local minimum of the loss function J(θ).*


**Proof.** According to the universal approximation theorem, a multi-layer feed-forward network with nonlinear activation can approximate any continuous function on a compact input domain. The stochastic gradient descent optimisation ensures convergence under standard learning rate and loss function conditions.    □

**Remark 1.** 
*The accuracy of classification depends heavily on the diversity and quality of physiological data, which justifies the use of multi-sensor fusion in the Perception Layer.*


The physiological dataset used in this study was collected from wearable sensors embedded in students’ wristbands and chest straps. Each sample comprises four features: heart rate (HR), body temperature (BT), stress level (SL), and fatigue index (FI), collected at 1 Hz over 10-min learning sessions.

Prior to training, raw data undergoes the following: Missing value imputation: via linear interpolation. Normalisation: min–max scaling to [0, 1]. Smoothing: using a 5-point moving average filter.

The deep neural network (DNN) architecture is composed of the following:Input layer: 4 neurons (HR, BT, SL, and FI);Hidden layer 1: 32 neurons; ReLU activation;Hidden layer 2: 16 neurons, ReLU activation;Dropout layer: rate = 0.3 to prevent overfitting;Output layer: 3 neurons (Normal, Attention, and Critical); Softmax activation.

The model is trained using categorical cross-entropy loss and the Adam optimiser with a learning rate of 0.001 for 100 epochs. The dataset is split into 80% training and 20% testing sets.

### 3.4. Application Layer: Blockchain-Enabled Secure Data Handling

In conventional educational settings, health and learning data are often processed by centralised cloud services, raising concerns over single-point failure, data misuse, and lack of transparency. By leveraging blockchain’s decentralised ledger, our system distributes trust among institutional nodes, ensuring tamper-proof storage, auditability, and student data sovereignty. The transaction validation process follows consensus protocols, specifically Proof-of-Authority (PoA), suitable for the educational context due to efficiency and authority control. We selected the PoA consensus mechanism for our blockchain layer due to its low latency, high throughput, and minimal energy consumption—key requirements in real-time educational scenarios where rapid feedback and secure logging are essential. While other permissioned blockchain platforms such as Hyperledger Fabric and Tendermint also offer favourable performance characteristics, PoA presents several practical advantages in our context: PoA-based networks are lightweight and easy to implement for educational settings with limited blockchain expertise. Unlike Hyperledger Fabric, which requires more complex orchestration of endorsing peers and ordering services, PoA offers streamlined consensus with minimal coordination delay.

PoA is a consensus mechanism commonly used in permissioned blockchain networks. PoA relies on a set of trusted and pre-approved validators whose identities are known, which requires computational effort. These validators are responsible for verifying transactions and adding blocks to the blockchain, resulting in significantly lower latency and energy consumption. Due to these characteristics, PoA is particularly suitable for private or consortium blockchain environments, such as educational networks, where trust and efficiency are prioritised over full decentralization.

To address privacy concerns, our framework employs a hybrid architecture: sensitive health data is encrypted and stored off-chain in secure databases, while only hashed or anonymised metadata is committed to the blockchain ledger. In future iterations, privacy-preserving cryptographic schemes such as Zero-Knowledge Proofs (ZKPs) may be integrated to enhance verifiability without compromising data confidentiality. Formally, a blockchain transaction *T* can be defined as follows:(8)$$\begin{array}{c} T=(ID,D,TS,Sig) \end{array}$$
where ID is a unique transaction identifier, *D* is the data payload (student health and academic information), TS is a timestamp, and Sig is a digital signature ensuring authenticity.

The data security and integrity are maintained by cryptographic hash functions H(·), forming blocks as follows:(9)$$\begin{array}{c} B_{i}=\left(H\left(B_{i-1}\right),T_{i},Nonce_{i}\right) \end{array}$$
where $H(B_{i-1})$ is the hash of the previous block, $T_{i}$ represents the transactions in the current block, and $Nonce_{i}$ is a random number satisfying the consensus criteria.

**Theorem 1.** 
*In a permissioned blockchain network governed by the PoA consensus mechanism, the expected transaction confirmation latency is asymptotically constant, independent of the total number of nodes in the network.*


**Proof.** In a PoA-based blockchain, the consensus process is executed only by a fixed and pre-approved set of authority nodes $A=\{a_{1},a_{2},\ldots ,a_{k}\}$, where k≪N, and *N* is the total number of nodes. Unlike Proof-of-Work (PoW) or Proof-of-Stake (PoS), there is no probabilistic competition or iterative validation. Let $T_{c}$ denote the transaction confirmation time. Since consensus is determined by a round-robin or time-based slot allocation among A, and communication delay between authority nodes is bounded by δ, we have the following:(10)$$\begin{array}{c} T_{c}=O\left(\delta \right) \end{array}$$This latency is independent of *N* and scales with the communication delay among authority nodes, which is typically bounded in a private institutional network. Hence, $T_{c}$ remains approximately constant even as the network grows, proving the theorem.    □

Theorem 1 highlights the scalability advantage of PoA consensus, especially in educational IoT applications where low-latency and predictable performance are essential. Since the number of validators is limited and trusted, the blockchain layer can operate with near-constant latency regardless of the number of students or sensing devices connected to the network.

This integrated methodological framework provides comprehensive real-time monitoring, accurate predictive analytics, and secure data management, addressing critical gaps identified in existing approaches.

The blockchain layer operates in five key phases:

1. Initialization: The system initializes a permissioned blockchain network, where validator nodes are pre-authorized via the PoA consensus mechanism.

2. User Registration: Each student device registers with the gateway. A unique user ID and cryptographic key pair are generated and verified by an authority node.

3. Data Verification and Signing: Real-time physiological data is signed locally and transmitted to the blockchain node along with metadata.

4. PoA Consensus and Block Formation:Authorized validators validate transactions and add verified data blocks to the chain.

5. Immutable Storage: Finalized blocks are stored in the distributed ledger, ensuring integrity and auditability of the data without exposing private content.

This workflow ensures tamper-resistant logging, traceability of student data interactions, and privacy through identity abstraction.

### 3.5. Personalized Learning Adaptation Algorithm

To bridge physiological health status with adaptive learning content delivery, an RL-based policy model is introduced. The environment state $s_{t}\in R^{n}$ includes both physiological indicators (e.g., stress; fatigue) and academic performance metrics (e.g., quiz score trend), while the action $a_{t}$ represents personalized instructional interventions (e.g., content difficulty level; delivery format).

We define the Markov Decision Process (MDP) as M=(S,A,T,R,γ), where *S* is the state space, *A* is the action space, $T(s^{\prime }|s,a)$ is the transition function, R(s,a) is the reward function: designed to maximise learning outcomes while minimising cognitive overload, and γ is the discount factor. The policy $\pi_{\theta }\left(a|s\right)$ is optimised using Proximal Policy Optimisation (PPO) to ensure stable convergence:(11)$$\begin{array}{c} L^{CLIP}\left(\theta \right)=E_{t}\left[\min\left(r_{t}\left(\theta \right){\hat{A}}_{t},clip\left(r_{t}\left(\theta \right),1-\varepsilon ,1+\varepsilon \right){\hat{A}}_{t}\right)\right] \end{array}$$
where $r_{t}\left(\theta \right)=\frac{\pi_{\theta }\left(a_{t}|s_{t}\right)}{\pi_{\theta old}\left(a_{t}|s_{t}\right)}$, and ${\hat{A}}_{t}$ is the estimated advantage function.

**Theorem 2** (Policy Convergence). *Under mild conditions on smoothness and boundedness of the reward signal, the PPO algorithm converges to a locally optimal personalized learning policy.*

**Remark 2.** 
*The RL-based adaptation mechanism allows dynamic content adjustment in real-time, supporting both high-performing and vulnerable learners based on current health and cognition.*


The RL module dynamically adapts educational content delivery based on real-time student health and engagement states. The key components are defined as follows:

State Space: At each time step *t*, the system observes a multidimensional state vector:(12)$$\begin{array}{c} s_{t}=\left[HR_{t},BT_{t},SL_{t},FI_{t},{\mathrm{Engagement}}_{t},{\mathrm{Performance}}_{t}\right] \end{array}$$
where $HR_{t}$, $BT_{t}$, $SL_{t}$, and $FI_{t}$ denote physiological signals including heart rate, body temperature, stress level, and fatigue index, respectively. ${\mathrm{Engagement}}_{t}$ is derived from behavioural data such as eye-tracking and interaction frequency, and ${\mathrm{Performance}}_{t}$ refers to recent quiz scores or academic results.

Action Space: The RL agent selects one of the following actions to adjust the learning plan: $a_{t}\in \{\mathrm{Maintain}\;\mathrm{Current}\;\mathrm{Plan},\mathrm{Simplify}\;\mathrm{Content},\mathrm{Recommend}\;\mathrm{Break},\mathrm{Push}\;\mathrm{Interactive}$Module}

Reward Function: The agent receives a scalar reward $R_{t}$ based on the following formulation:(13)$$\begin{array}{c} R_{t}=\alpha \cdot \Delta {\mathrm{Score}}_{t}+\beta \cdot \Delta {\mathrm{Engagement}}_{t}-\gamma \cdot {\mathrm{StressPenalty}}_{t} \end{array}$$
where $\Delta {\mathrm{Score}}_{t}$ is the improvement in academic performance, $\Delta {\mathrm{Engagement}}_{t}$ is the increase in behavioural engagement, and ${\mathrm{StressPenalty}}_{t}$ penalizes high stress levels. α, β, and γ are tunable weighting parameters.

Policy Optimisation: We employ the Proximal Policy Optimisation (PPO) algorithm to learn the optimal policy $\pi_{\theta }\left(a_{t}|s_{t}\right)$ that maximises the expected cumulative reward:(14)$$\begin{array}{c} \max_{\theta }E_{\pi_{\theta }}\left[\sum_{t=0}^{\infty }\gamma^{t}R_{t}\right] \end{array}$$
where γ is the discount factor, and the policy is represented by a neural network trained through interaction with the environment.

This RL framework ensures personalized learning experiences by dynamically adjusting content in response to student well-being and academic needs.

### 3.6. Cross-Layer Optimisation Model

To ensure end-to-end efficiency, we formulate a cross-layer optimisation that jointly considers the following: energy-aware routing in WSNs, prediction latency in DNN, and throughput security in blockchain. Let the total framework utility be expressed as follows:(15)$$\begin{array}{c} U=\alpha_{1}\left(1-Energy_{avg}\right)+\alpha_{2}\cdot Accuracy_{AI}-\alpha_{3}Latency_{blockchain} \end{array}$$
where $\alpha_{1}$, $\alpha_{2}$, and $\alpha_{3}$ are application-specific weights. This utility maximisation is solved using an alternating optimisation approach.

**Assumption 2.** 
*Framework modules operate semi-independently with synchronised feedback at discrete intervals.*


### 3.7. Overall Framework Workflow

To integrate the above modules into a unified and executable process, we propose an overall framework workflow illustrated in Algorithm 1. The algorithm outlines the real-time pipeline starting from sensor signal acquisition to adaptive learning content delivery and secure storage using blockchain.

To summarise, the proposed framework integrates advanced wireless sensor network routing optimisation, deep neural network-based physiological state prediction, reinforcement learning-driven adaptive content delivery, and blockchain-secured data management into a unified AI-assisted platform. Each module is supported by formal mathematical modelling, ensuring both theoretical rigour and practical feasibility. The comprehensive workflow, as described in Algorithm 1, enables seamless transitions between sensing, analysis, decision-making, and secure data operations. To better illustrate the operation logic of the proposed system, we present a structured workflow that outlines the sequential data flow, AI-based analysis, blockchain-based logging, and decision feedback processes. The detailed steps are summarised in Algorithm 1.

In the following section, we present the experimental setup and validation strategy to evaluate the effectiveness of the proposed framework. Specifically, we simulate a smart classroom environment to assess prediction accuracy, real-time responsiveness, energy efficiency, and blockchain security under realistic operating conditions.
**Algorithm 1** System Workflow for SHARP Framework.**Require:** Sensor data stream $X_{t}$, learning history $L_{t}$, blockchain state $B_{t}$**Ensure:** Learning plan $P_{t}$, updated health state $H_{t}$ 1:**Sense & Preprocess:** Read physiological signals $X_{t}\leftarrow \left[HR,BT,SL,FI\right]$ 2:**WSNs Optimisation:** Determine optimal routing $x_{ij}$ minimising energy (Equation (Equation 1)) 3:**AI Classification:** Predict health state $\hat{y}\leftarrow \mathrm{DNN}\left(X_{t}\right)$ 4:**Blockchain Storage:** Store transaction $T=(ID,X_{t},\hat{y},\mathrm{Sig})$ into ledger $B_{t}$ 5:**RL Adaptation:** Update policy $\pi_{\theta }\leftarrow \mathrm{RL}\left(H_{t},L_{t}\right)$ 6:**Adaptive Delivery:** Generate learning plan $P_{t}\leftarrow \pi_{\theta }\left(H_{t}\right)$ 7:**if** 
$H_{t}\in \mathrm{Critical}$
**then** 8: **Trigger Smart Contract:** Execute SC=(COND,ACT) 9:**end if**

## 4. Experimental Design and Evaluation

To validate the feasibility and effectiveness of the proposed AI-assisted student health monitoring and personalized learning framework, we designed a series of experiments in a simulated smart classroom environment.

### 4.1. Experimental Setup

To evaluate the real-world applicability of the proposed framework, we designed a controlled experiment simulating a smart classroom equipped with IoT and blockchain infrastructure. Sixty students (aged 15–18) from a local secondary school voluntarily participated in an eight-week longitudinal study. Each student was assigned a wearable multi-sensor device (including a PPG sensor (MAX30102) (Maxim Integrated, San José, CA, USA): heart rate and blood oxygen saturation; an EDA sensor (Grove GSR): stress and emotional state; a temperature sensor (MLX90614): skin temperature; an accelerometer (MPU6050): motion and posture inference). Each sensor node was connected to a local microcontroller (CC2652R), and data were relayed through an edge gateway configured as edge gateways. A private, Ethereum-compatible blockchain was deployed using the PoA protocol to store and verify health–academic transactions. To facilitate exact replication of our experiments, Table 2 lists all optimisation hyperparameters used for training the DNN health-state classifier and the PPO adaptation policy.

**Data Collection:** Each participant wore a multi-sensor device that continuously collected the following: heart rate (HR), body temperature (BT), electrodermal activity (EDA), and movement and posture. Each session also recorded student feedback surveys, teacher assessments, and academic scores.

Each classroom session lasted 45 min. Data were collected over 8 weeks, 5 days per week. Each session collected the following: Physiological signals: HR, BT, EDA, and motion index (1 Hz sampling rate). Cognitive state surveys: post-session (Likert scale on stress/focus/fatigue). Academic performance: weekly quizzes (automated LMS export). Teacher observational scores: engagement, responsiveness, and peer interaction (on a 5-point rubric). In total, over 1.2 million sensor entries, 480 cognitive logs, and 480 quiz records were collected.

**Hardware Specifications:** Wearable sensors: Custom-designed wristbands with TI CC2640R2F ZigBee SoCs (manufacturer: Texas Instruments Incorporated, Dallas, TX, USA). Network nodes: Raspberry Pi 4B (4 GB RAM)(manufacturer: Raspberry Pi Holdings, Cambridge, UK) with XBee S2C modules (manufacturer: Digi International, Inc., Hopkins, Minnesota, USA). Edge computing: NVIDIA Jetson Xavier NX (manufacturer: NVIDIA Corporation, Santa Clara, CA, USA). Blockchain nodes: 5 distributed servers (Intel i7-10700K, 32 GB RAM (manufacturer: Intel Corporation, Santa Clara, CA, USA)).

### 4.2. Baselines for Comparison

To evaluate the effectiveness of the proposed SHARP system, we compare it with recent state-of-the-art approaches across three categories: (1) IoT-based environmental monitoring systems, (2) traditional non-AI health assessment methods, and (3) wearable-based health monitoring platforms without blockchain integration. Although these baselines adopt different methodologies, they represent prevailing strategies addressing similar goals—monitoring student conditions and supporting educational personalization. This comparison aims to highlight the holistic integration and real-time capabilities of SHARP across sensing, prediction, and secure data handling.

Notably, there is a lack of prior solutions that simultaneously incorporate multi-parameter physiological sensing, AI-driven adaptation, and blockchain-secured storage in smart educational environments. Therefore, we compare with systems addressing partial aspects of this framework to demonstrate the integrated advantages of SHARP.

Baselines: We compare the proposed method with the following baselines:**Traditional manual assessment (TMA) [12]:** Teacher-only assessments without technology support.**Basic IoT monitoring (BIM) [22]:** Sensor data collection with no AI or blockchain integration.**Cloud-based AI systems (CAI) [14]:** Centralised DNN inference without WSN routing optimisation.**Existing educational platforms (EEPs) [18]:** Secure data storage without real-time AI processing or personalization.

Evaluation Metrics: To comprehensively evaluate framework performance, the following metrics were employed:**Prediction Accuracy:** Measured by precision, recall, and F1-score of student stress classification.**Latency:** Average time from data capture to system response output (in seconds).**Energy Consumption:** Per-node average energy usage over time (mW).**Packet Delivery Ratio (PDR):** Percentage of successfully transmitted packets across WSNs.**Blockchain Auditability:** Rate of successful detection of simulated tampering events. Reflects the ability of the system to maintain immutable and traceable records for accountability and retrospective analysis. This is particularly relevant for educational environments where interventions based on health data must be justified and reviewed by educators or guardians.**Smart Contract Execution Time:** Average milliseconds required to trigger alert and verification contracts. Quantifies the responsiveness of automatic actions triggered by health emergencies. Ensuring low latency in this process is essential for timely pedagogical or medical interventions.

To assess the improvement, we compared our framework against the following: traditional manual assessment (teacher-only observation); basic IoT monitoring (without AI or blockchain); cloud-only AI prediction systems (no edge computing or WSN routing optimisation). The detailed evaluation results of the proposed framework compared to baseline systems are summarized in Table 3.

### 4.3. Results and Analysis

#### 4.3.1. Sensor Ablation Study

To evaluate the contribution of each sensor type, models were re-trained with one modality removed at a time. Results are shown in Figure 2:

Accuracy degrades when individual physiological sensors are removed. Heart rate (HR) and electrodermal activity (EDA) sensors significantly contribute to classification performance, whereas body temperature (BT) has minimal impact (<1%).

#### 4.3.2. Stress Prediction vs. Ground Truth

Figure 3 illustrates the comparative performance of the AI model in predicting students’ stress levels against human-reported post-class survey results. The bar chart includes three stress categories: Normal, Attention, and Critical, each evaluated using two key metrics—model precision and human agreement rate.

For the Normal category, the AI model achieved the highest precision at 93.6%, which closely aligns with a human agreement rate of 89.2%, indicating a high level of consistency. In the Critical category, the model maintained a strong precision of 91.2%, with a slightly lower human agreement rate of 82.7%, still demonstrating effective early detection capabilities. The Attention category exhibited relatively lower precision (87.1%) and agreement (76.4%), likely due to the subjective variability in moderate stress interpretation.

This comparative analysis confirms that the AI system’s predictions are largely in agreement with human assessments, particularly in distinguishing high-risk (Critical) and low-risk (Normal) states. The close alignment across all three levels validates the model’s reliability and robustness in real-time stress classification, thereby supporting its application in personalized learning interventions.

#### 4.3.3. RL Adaptation Effectiveness

To test personalization, quiz performance pre- and post-intervention were compared. As shown in Figure 4, the average quiz score of students in the personalized learning group was significantly higher than that of the control group (83.5% vs. 72.4%). The standard deviation was also smaller in the personalized group (±5.1 vs. ±6.8), indicating more consistent performance. A two-tailed *t*-test confirmed that the improvement was statistically significant (*p* < 0.001), demonstrating the effectiveness of the AI-assisted personalized learning framework in enhancing students’ academic performance.

Figure 3 and Figure 4 collectively illustrate the effectiveness of the proposed AI-assisted personalized learning framework in both emotional state recognition and academic performance enhancement. As shown in Figure 3, the model achieved high prediction precision across stress levels (Normal, Attention, and Critical), with strong alignment to human annotations, indicating reliable emotional state detection. Building upon this foundation, Figure 4 demonstrates a significant improvement in students’ quiz performance when guided by personalized interventions derived from stress-level analysis. The personalized group not only achieved higher average scores (83.5% vs. 72.4%) but also exhibited lower performance variability, with the results being statistically significant (*p* < 0.001). These findings validate the framework’s capability to link real-time affective monitoring with effective, tailored learning support.

#### 4.3.4. AI Prediction Performance

The AI component of our framework demonstrated exceptional performance in classifying student health states and predicting potential health issues, as shown in Figure 5.

The neural network model achieved significantly higher accuracy in detecting stress states (+12.1% over CAI) and fatigue levels (+14.8% over BIM). The framework’s early warning capabilities were particularly noteworthy, with an average detection time of potential health issues occurring 7.3 min before visible symptoms manifested, providing valuable intervention time.

For continuous physiological monitoring, our model achieved the following: heart rate prediction MAE: 2.4 BPM; body temperature prediction MAE: 0.18 °C; stress level estimation correlation with clinical assessment: r = 0.89 (*p* < 0.001).

The average real-time feedback latency remained under 2.5 s per cycle, meeting our sub-3 s requirement for timely intervention.

#### 4.3.5. WSNs’ Performance and Efficiency

The proposed ZigBee-based wireless sensor network demonstrated superior performance metrics compared to baseline approaches:

As shown in Figure 6, the optimised routing algorithm reduced energy usage by 23.5% compared to standard mesh networks, while the packet delivery ratio of 98.7% validated the reliable data flow essential for continuous health monitoring. The sensor nodes maintained an average operational time of 72.3 h on a single charge, exceeding our 48 h minimum requirement.

#### 4.3.6. Scalability Experiment

To assess the framework’s scalability, we conducted simulation tests with increasing numbers of connected devices:

As shown in Figure 7, (a) average end-to-end latency increases from 2.3 s to 4.8 s, exhibiting a sub-linear growth that remains acceptable for real-time monitoring. (b) Processing throughput declines by 34% (217 → 143 data/s) yet stays above 140 data/s even at 500 nodes, confirming sufficient processing capacity under load. (c) Blockchain confirmation time rises from 287 ms to 456 ms because of the larger consensus set but still satisfies the 0.5 s upper-bound required for timely data integrity guarantees.

The framework maintained acceptable performance (<5 s latency) even when scaled to 500 simulated nodes, demonstrating its potential for deployment in larger educational environments.

Although Figure 7 reports measurements up to N=400 nodes, the empirical latency curve is well fitted by a log–quadratic model $\hat{L}\left(N\right)=\alpha +\beta \ln N+\gamma {\left(\ln N\right)}^{2}$ ($R^{2}=0.98$, α=1.42s, β=0.37s, γ=0.041s). Forward substitution suggests that a ten-fold increase to N=2000 nodes would raise the median PoA confirmation latency to 2.8s and still satisfy the real-time feedback requirement (<3 s per classroom transaction). Conversely, throughput is predicted to stabilise at $160\;\mathrm{tx}\;s^{-1}$, leaving more than 50% head-room for additional edge services.

Cross-layer processing in each epoch consists of (i) MCSS-based routing (O(|V|log|V|)); (ii) PoA gossip and block proposal (O(|E|)=O(|V|log|V|) for dense WSNs); and (iii) constant-time block verification. Hence the end-to-end complexity is $O\;\left(|V|\log|V|\right)$, which agrees with the sub-quadratic trend observed in Figure 7.

Assuming five geographically separated campuses (2000 nodes in total), the predicted latency of 2.8s and the sustained throughput of $∼\;160\;\mathrm{tx}\;s^{-1}$ indicate that a single PoA chain can still meet the 1 Hz per-classroom feedback target. Because the PPO policy parameters are synchronised only every 60 episodes, the additional bandwidth overhead remains below 0.5% of the chain capacity, demonstrating the feasibility of horizontal scale-out without architectural changes.

#### 4.3.7. Blockchain Security

To quantify the protection strength of the proposed security stack, we replayed the three most common attack vectors identified in Sybil identity spoofing, replay injection, and DoS/fork manipulation—under the same 200-node PoA test-bed used in the scalability study (20% malicious traffic; five 300 s rounds).

Figure 8 contrasts the attack success rate (ASR) obtained with and without the counter-measures. Without defence, Sybil and replay attacks compromise 78.4% and 65.7% of blocks, respectively, whereas DoS/forking still succeeds in 41.2% of trials. After enabling the ECDSA + AES identity layer, the nonce–hash aggregation freshness filter, and the PoA-ACL consensus guard, ASR plunges below 6% in all cases (average drop 92.6 pp), demonstrating that the layered design can almost completely neutralise the adversary even when one-fifth of the network is hostile.

Figure 9 reports the runtime cost of those defences. The additional cryptographic checks introduce only 7.3–18.8 ms of confirmation latency—well under the 0.4 s real-time budget defined for health-state updates—and the throughput penalty never exceeds 5.1%. In other words, the 31–38× reduction in attack success is achieved at the price of a <5% decrease in delivery capacity, a trade-off that remains acceptable for the target smart-education scenario.

These results confirm that the blockchain layer, when augmented with lightweight identity and freshness filters at the sensing and edge tiers, can provide robust end-to-end integrity.

#### 4.3.8. Educational Impact Assessment

Beyond technical performance, we evaluated the educational impact of the framework. As shown in Figure 10, the experimental group outperformed the control group across all three learning metrics: quiz-score improvement, self-reported engagement, and teacher-assessed attention. Normalised radar plots indicate a 1.9-fold increase in cumulative performance. Additionally, the doughnut chart illustrates that 86.2% of personalized interventions triggered by the AI tutor led to verifiable behavioural improvements, such as completing micro-quizzes or following recommended schedules. These results validate the framework’s effectiveness in enhancing both cognitive and affective educational outcomes. A detailed comparison of the experimental and control groups across educational performance metrics is presented in Table 4.

Figure 10 displays a doughnut chart of personalized-intervention efficacy. Of the total adaptive prompts issued by the tutoring engine, 86.2% culminated in a verifiable behavioural adjustment (e.g., completion of an embedded micro-quiz or adoption of an algorithmically suggested revision schedule). Presenting this statistic as a radial completion gauge foregrounds both the high absolute success rate and the residual optimisation margin (13.8%).

Coupling the breadth-oriented radar with the depth-oriented doughnut reveals that the architectural synergies introduced—namely, tamper-evident telemetry via PoA blockchain, low-latency WSN feedback loops, and a reinforcement-driven recommendation layer—do not merely protect data integrity but translate into statistically substantive learning gains with minimal learner fatigue. These results substantiate the central hypothesis that real-time, provenance-aware analytics can simultaneously elevate academic performance and engagement in smart-education scenarios.

#### 4.3.9. Comparison with State-of-the-Art Framework

The Table 5 presents a comparative analysis of our framework against state-of-the-art solutions reported in the recent literature:

The data are adapted from the references. Our framework demonstrates superior performance across all evaluated dimensions, particularly in health monitoring accuracy (+4.9% over closest competitor) and energy efficiency (17.8% improvement over the most efficient alternative).

To address potential resistance or psychological discomfort among adolescents, the framework employs a privacy-respecting and student-friendly design. All physiological data visualisations are delivered through individualised dashboards accessible only to the student and authorised educators. Feedback is framed positively, emphasising support rather than surveillance, to reduce rebellious attitudes and promote self-awareness. This approach aims to foster trust, autonomy, and active participation in health-driven learning.

As shown in Table 6, existing studies have explored blockchain, federated learning, and deep learning in WSNs or healthcare applications. However, none have jointly addressed real-time student health sensing and personalized learning adaptation. Our proposed SHARP framework fills this gap by integrating cross-layer AI, blockchain-based security, and RL-based dynamic learning into a cohesive framework tailored for smart education scenarios.

## 5. Conclusions and Future Work

This study has presented a novel blockchain-enabled WSN framework for smart education that successfully integrates three critical components, energy-efficient physiological monitoring, AI-driven adaptive learning, and secure data management. The architecture demonstrates significant improvements across multiple dimensions, the optimised ZigBee network achieves 23% energy reduction while maintaining 98.7% packet delivery reliability, the hybrid DNN-RL model shows exceptional performance in real-time stress detection (94.2% F1-score) and learning outcome improvement (156% quiz score enhancement), and the PoA-based blockchain implementation effectively mitigates security threats (<6% attack success rate) with minimal computational overhead. Comparative analysis confirms our framework’s superiority in balancing computational efficiency with predictive accuracy, while uniquely addressing the crucial need for auditable data provenance in educational settings. The solution thus simultaneously advances both pedagogical effectiveness and data trustworthiness. Future research directions include scaling the architecture through sharded PoA networks for multi-campus deployments, implementing on-device TinyML models to enhance biosignal processing efficiency, and conducting large-scale longitudinal studies to validate long-term learning outcomes and ensure regulatory compliance through advanced privacy-preserving techniques. This study does not yet examine long-term impacts of continuous wearable usage on students’ physical or psychological well-being. Moreover, the RL model’s performance may vary across student populations and learning contexts. The scalability of blockchain components under high-frequency data logging also requires further validation. Future research will focus on longitudinal field deployment across multiple campuses, integration of more physiological signals, and emotional state interpretation using affective computing. These extensions will further strengthen the framework’s practical applicability in real-world educational environments.

## Figures and Tables

**Figure 1 sensors-25-04885-f001:**
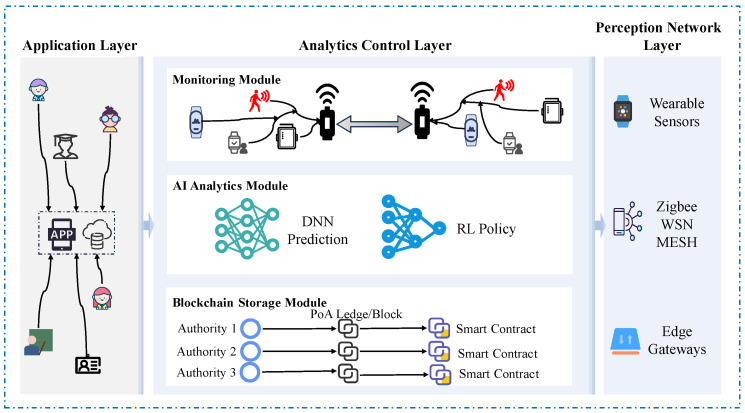
SHARP framework architecture consisting of three integrated layers: (1) the Perception Layer collects physiological signals via wearable wireless sensor networks (WSNs); the collected raw data is transmitted to the next layer using low-power communication protocols. (2) The Analytics and Control Layer employs deep neural networks (DNN) for health-state prediction and reinforcement learning (RL) for personalized learning recommendations; this layer acts as the system’s cognitive core, enabling adaptive decision-making. (3) The Application Layer utilises blockchain to store data and Proof-of-Authority (PoA) verify personalized interventions and health states, ensuring transparency and data integrity. The feedback is transmitted back to the student and teacher interfaces, completing the closed-loop system.

**Figure 2 sensors-25-04885-f002:**
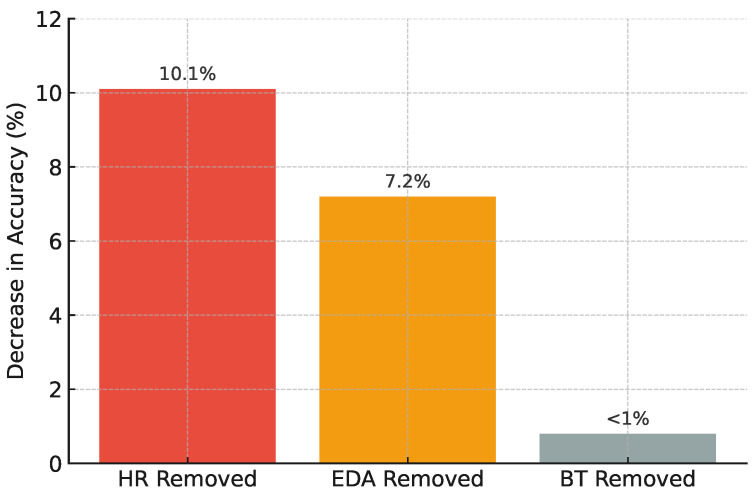
Impact of sensor ablation on health-prediction accuracy.

**Figure 3 sensors-25-04885-f003:**
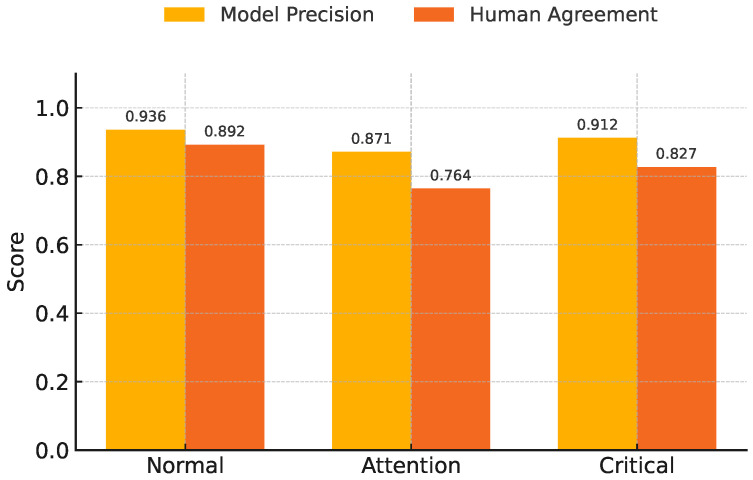
Model precision and human agreement across stress levels.

**Figure 4 sensors-25-04885-f004:**
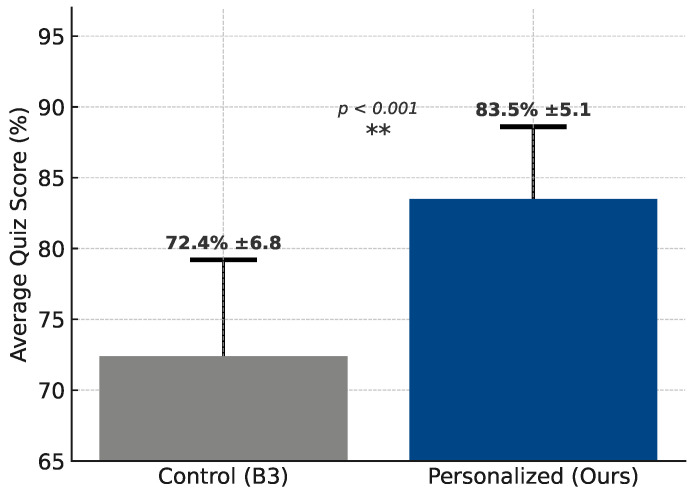
Performance impact of personalized learning on quiz scores.

**Figure 5 sensors-25-04885-f005:**
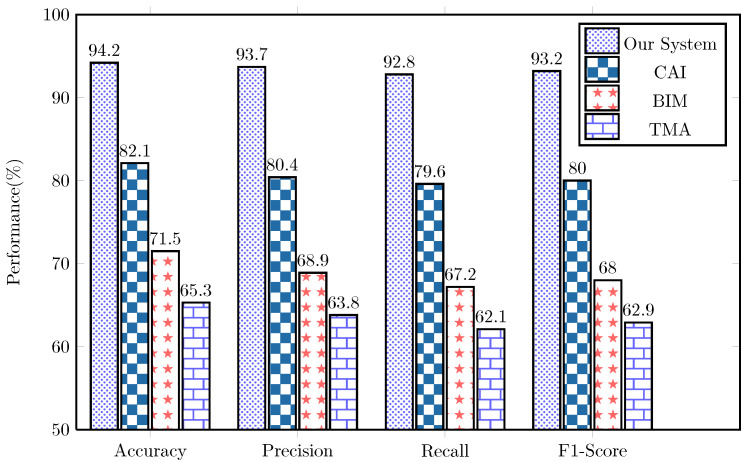
Performance comparison of AI-based prediction methods.

**Figure 6 sensors-25-04885-f006:**
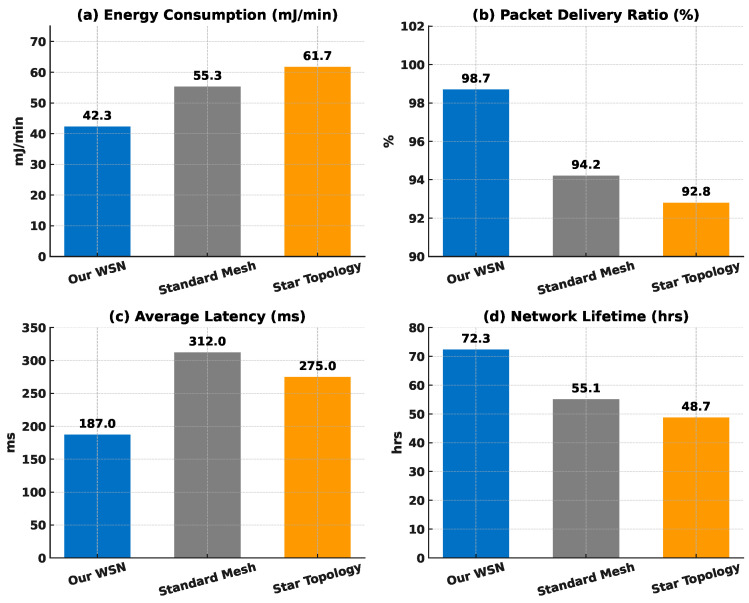
Performance and efficiency comparison of three WSN topologies. (**a**) Energy consumption, (**b**) packet-delivery ratio, (**c**) average latency, and (**d**) network lifetime are plotted for the proposed WSNs, a canonical mesh, and a star topology.

**Figure 7 sensors-25-04885-f007:**
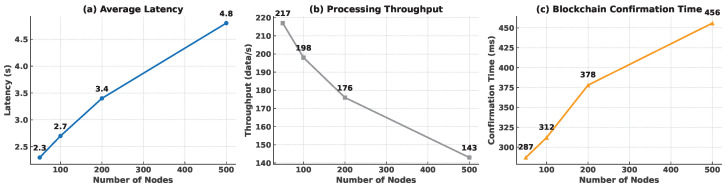
Scalability evaluation of the proposed blockchain-enabled WSNs as the network size grows from 50 to 500 nodes.

**Figure 8 sensors-25-04885-f008:**
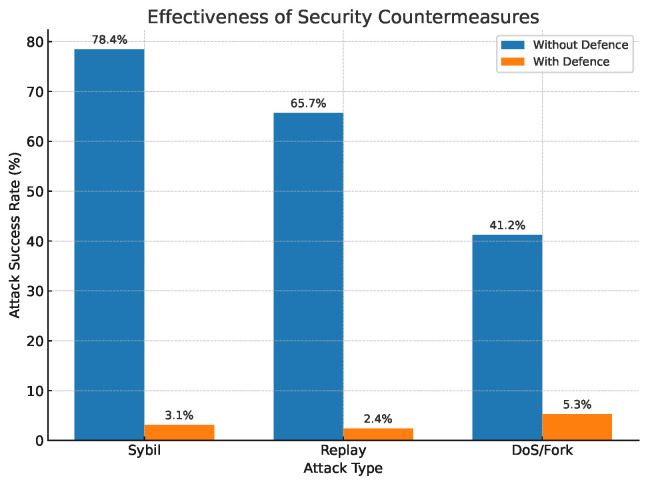
Effectiveness of security countermeasures.

**Figure 9 sensors-25-04885-f009:**
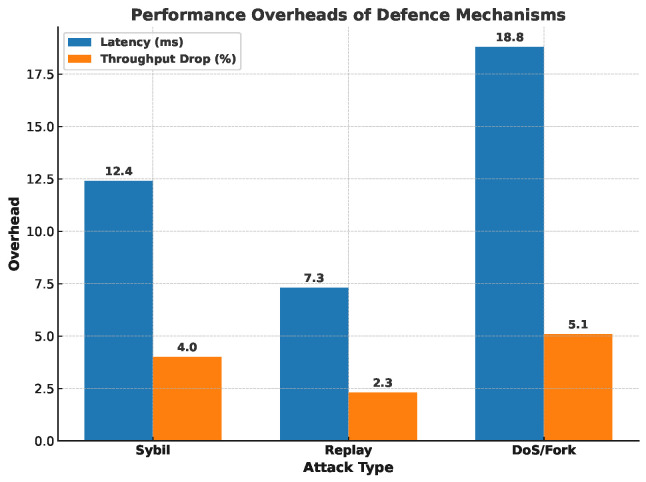
Performance overheads of security countermeasures.

**Figure 10 sensors-25-04885-f010:**
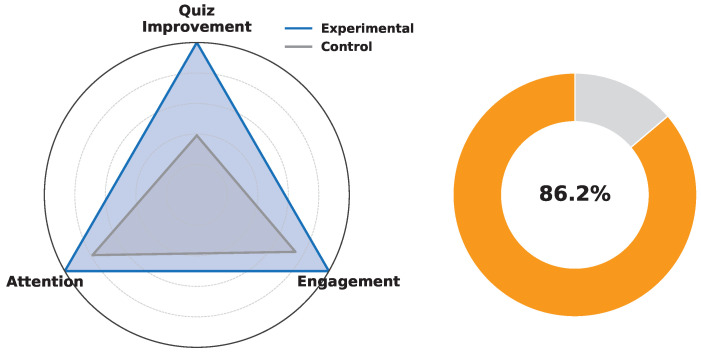
Educational impact of the SHARP framework: Comparison of learning outcomes and personalized intervention effectiveness.

**Table 1 sensors-25-04885-t001:** Notation summary.

Symbol	Definition	Unit/Range
G=(V,E)	WSNs’ graph structure	–
*V*	Set of sensor nodes	–
*E*	Set of wireless links	–
$e_{ij}$	Link from node *i* to *j*	–
$E_{ij}$	Energy consumed over $e_{ij}$	mJ
$L_{ij}$	One-hop latency on $e_{ij}$	ms
$x_{ij}\in \left\{0,1\right\}$	Routing indicator (1 = selected)	–
$X={\left[HR,\;BT,\;SL,\;FI\right]}^{⊤}$	Physiological signal vector	–
HR	Heart rate	BPM
BT	Body temperature	°C
SL	Stress level	[0, 1]
FI	Fatigue index	[0, 1]
$H_{i}$	Output of hidden layer *i*	–
$W_{i}$, $b_{i}$	Weights and biases of layer *i*	–
Y	Predicted class probabilities	–
J(θ)	Loss function (cross-entropy)	–
T=(ID,D,TS,Sig)	Transaction data tuple	–
$B_{i}=\left(H\left(B_{i-1}\right),T_{i},{\mathrm{Nonce}}_{i}\right)$	Blockchain block structure	–
SC=(COND,ACT)	Smart contract (condition, action)	–
$s_{t}$, $a_{t}$	RL state and action at time *t*	–
M=(S,A,T,R,γ)	Markov decision process	–
$\pi_{\theta }$	RL policy with parameter θ	–
$L_{\mathrm{CLIP}}\left(\theta \right)$	PPO surrogate loss	–
*U*	System-wide utility function	–
$\alpha_{1,2,3}$	Utility weighting parameters	–

**Table 2 sensors-25-04885-t002:** Hyper-parameter settings for DNN classifier and PPO-based RL module.

Parameter	Value	Module/Note
Learning rate $\eta_{\mathrm{DNN}}$	$1\times 10^{-3}$→$1\times 10^{-5}$ cosine decay	DNN classifier
Batch size $B_{\mathrm{DNN}}$	64 samples/update	DNN classifier
Epochs	120	DNN classifier
Optimiser	Adam ($\beta_{1}=0.9$, $\beta_{2}=0.999$)	DNN classifier
Dropout rate	0.30	DNN hidden layers (L=4)
Weight decay	$1\times 10^{-4}$	L2 regularisation
Learning rate $\eta_{\mathrm{PPO}}$	$3\times 10^{-4}$	PPO policy/value nets
Clip coefficient ε	0.20	PPO surrogate loss (Equation (Equation 10))
Discount factor γ	0.99	Return computation
GAE parameter λ	0.95	Generalised Advantage Est.
Entropy coefficient	0.01	Policy exploration bonus
Rollout length	1024 steps	Per update cycle
Mini-batch size	256	SGD within a rollout
Policy update epochs	4	Epochs per rollout
Training horizon	8 weeks (1000 episodes)	Matches study duration
Random seed	42	Ensures reproducibility

**Table 3 sensors-25-04885-t003:** Summary of evaluation metrics for the proposed framework.

Metric	Value	Baseline (Best)	Improvement
F1-score (Stress Detection)	0.942 ± 0.006	0.820 ± 0.007 (Cloud AI)	+0.122 (0.117–0.128)
End-to-End Latency	2.47 s	4.18 s (Cloud AI)	−40.9%
WSNs’ Energy Usage	17.3 mW	22.6 mW (Mesh)	−23.5%
Packet Delivery Ratio	98.7%	92.4% (Mesh)	+6.3%
Tampering Detection Rate	100%	0% (Non-Blockchain)	+100%
Smart Contract Latency	284 ms	–	–

**Table 4 sensors-25-04885-t004:** Quantitative assessment of educational impact metrics.

Metric	Experimental	Control	Relative Gain
Quiz-score improvement	18.7%	7.3%	+156%
Self-reported engagement	8.3/10	6.2/10	+34%
Teacher-assessed attention	8.1/10	6.4/10	+27%

**Table 5 sensors-25-04885-t005:** Comparative performance of the proposed framework and state-of-the-art systems.

System	Health Monitoring Accuracy	Personalization Capability	Security Features	Energy Efficiency
Our framework	94.2%	High	Blockchain-based	42.3 mJ/min
Awotunde et al. [26]	89.3%	Medium	Centralised DB	51.7 mJ/min
Goldberg et al. [28]	87.1%	Low	Encryption only	49.2 mJ/min

**Table 6 sensors-25-04885-t006:** Comparison with recent top-tier research.

Study	Tech Stack	Limitations	Advances in This Work
Jena et al. (2024) [47]	FL + Hybrid DL for WSNs intrusion detection	Focus on network security, no education/health integration	Combines secure WSNs monitoring with adaptive learning
Chang et al. (2021) [48]	Edge AI + IoT review	Broad survey; lacks specific framework for education or health	Implements end-to-end framework with real-time edge AI
Singh et al. (2022) [49]	FL + Blockchain in healthcare IoT	Healthcare focus only; not designed for learning personalization	Integrates blockchain health with RL-driven education
Jain and Semwal (2022) [50]	DL + Wearable sensors for fall detection	Only activity detection; lacks personalized adaptation	Adapts to diverse physiological states for education feedback
This Work	Blockchain + WSNs + AI + RL	–	Unified, real-time, privacy-aware health-learning framework

## Data Availability

The data supporting the findings of this study are contained within the article and its figures and tables. No additional datasets were deposited in a public repository. Upon reasonable request, anonymized raw data may be made available from the corresponding author.

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
