# Peer review of "SHARP: Blockchain-Powered WSNs for Real-Time Student Health Monitoring and Personalized Learning"

_sensors, 2025, doi:10.3390/s25164885_

Round 1
Reviewer 1 Report
Comments and Suggestions for Authors
This paper proposes the SHARP framework, which enables real-time student health monitoring and personalized learning in smart educational environments through blockchain-enhanced WSNs. It is well organized and it is interesting. However, the manuscript has some issues needed to be addressed . Below are detailed comments and suggestions for improving your work.
In lines1-18. The abstract should present the current state of research and the existing problems in this type of research.
The content of the abstract should explicitly highlight the advantages of the proposed method compared to previous studies in the same field, to better emphasize its innovative value.
The titles of some of the images in the text are too lengthy (Fig 10), which may hinder readers from quickly grasping their core content. It is recommended to simplify the titles and incorporate the detailed descriptions into the main text for clearer presentation.
In lines 22-37, the article summarizes current issues to be addressed in this research direction, but it overlooks several important aspects that require discussion, including but not limited to:
1) whether this method of using wireless sensors requires students to wear them for a long tie, and if so, whether it will cause harm to students' health, including physical damage caused by wearing items for a long time, and whether carrying such electronic devices for a long time will cause physical damage. For example, in Jinhua, Zhejiang, China in 2019, a primary school used a "surveillance headband", which caused concern among parents and society. 2: Although the initial intention of using these devices is good, it can easily lead to adolescent rebellious psychology, which in turn can lead to unhealthy psychological states. Does the device designed in the article have a method (such as personalized display) to reduce students' possible rebellious and resistant psychology. 3: Although there are currently no corresponding laws and regulations, some have pointed out that such behavior to some extent violates students' freedom rights. Is there a reason why most of these technologies are applied in the medical field rather than the educational field?
There are abnormal blank spaces in the text (page 6), which should be checked and corrected to ensure the standardization of the document format.
Additionally, the work should be compared with more relevant studies recently published. It is suggested to supplement comparisons with the latest literature to strengthen the persuasiveness of the innovation demonstration.
Author Response
Comments 1: In lines1-18. The abstract should present the current state of research and the existing problems in this type of research.
Response 1: Thank you for your valuable suggestion. We have revised the abstract to explicitly include the current state of the art and the key limitations in existing research, such as fragmented integration of health monitoring and learning systems, insufficient prediction accuracy, and unresolved privacy issues. These gaps now motivate the development of our proposed framework. The updated abstract better contextualises our contributions.
Comments 2: The content of the abstract should explicitly highlight the advantages of the proposed method compared to previous studies in the same field, to better emphasise its innovative value.
Response 2: We appreciate the reviewer’s insightful suggestion. In the revised abstract (Lines 1–21), we have explicitly added a comparative statement that highlights the advantages of our proposed SHARP framework over existing methods. Specifically, we emphasise its unique integration of multi-parameter physiological monitoring, AI-driven adaptation, and blockchain-based data security, which together lead to significantly improved accuracy, energy efficiency, and privacy compared to prior IoT-based educational systems. This revision clarifies the innovative value and novelty of our work.
Comments 3: The titles of some of the images in the text are too lengthy (Fig 10), which may hinder readers from quickly grasping their core content. It is recommended to simplify the titles and incorporate the detailed descriptions into the main text for clearer presentation.
Response 3: Thank you for the suggestion. We have revised the title of Figure 10 to make it more concise and reader-friendly:“Figure 10. Educational impact of the SHARP system: Comparison of learning outcomes and personalised intervention effectiveness.”
The explanatory details previously in the caption have been moved into the main text to improve clarity and maintain a clean visual layout. We believe this revision enhances both readability and presentation quality.
Comments 4: In lines 22-37, the article summarizes current issues to be addressed in this research direction, but it overlooks several important aspects that require discussion, including but not limited to: 1. whether this method of using wireless sensors requires students to wear them for a long tie, and if so, whether it will cause harm to students' health, including physical damage caused by wearing items for a long time, and whether carrying such electronic devices for a long time will cause physical damage. For example, in Jinhua, Zhejiang, China in 2019, a primary school used a "surveillance headband", which caused concern among parents and society.
Response 4: We thank the reviewer for raising this important and often overlooked point. In response, we have added a new paragraph to the Introduction section discussing potential health and ethical concerns related to long-term wearable sensor usage in educational settings. We specifically address the risks of physical discomfort, long-term device exposure, and social controversy, referencing the “surveillance headband” incident in Jinhua, China (2019). This addition enhances the contextual grounding of our work and underscores the importance of ethical considerations in smart education deployments involving minors.
Comments 5: Although the initial intention of using these devices is good, it can easily lead to adolescent rebellious psychology, which in turn can lead to unhealthy psychological states. Does the device designed in the article have a method (such as personalised display) to reduce students' possible rebellious and resistant psychology.
Response 5: Thank you for raising this important point regarding students' psychological acceptance. We have added a paragraph(At the end of "Experimental Design and Evaluation") explaining that the SHARP system incorporates privacy-preserving, student-centred display mechanisms to minimise rebellious psychology. By offering individualised dashboards with positive framing, the system promotes student autonomy and trust, reducing resistance and encouraging healthy engagement with the technology.
Comments 6: Although there are currently no corresponding laws and regulations, some have pointed out that such behavior to some extent violates students' freedom rights. Is there a reason why most of these technologies are applied in the medical field rather than the educational field?
Response 6: We appreciate the reviewer’s thoughtful observation regarding legal and ethical considerations. We have added a concise explanation (In the middle of the introduction) discussing why physiological monitoring technologies, while common in healthcare, raise different concerns in education. The SHARP framework addresses these concerns by ensuring voluntary participation, informed consent, and student data ownership. Unlike clinical systems, our design prioritises student empowerment and supportive intervention, aligning with ethical best practices despite the absence of specific educational regulations.
Comments 7: There are abnormal blank spaces in the text (page 6), which should be checked and corrected to ensure the standardisation of the document format.
Response 7: Thank you for pointing out the formatting inconsistency. We have carefully reviewed the document and corrected the abnormal blank spaces on page 6. The issue was caused by forced float positioning and spacing inconsistencies around Table 1 and mathematical expressions. We have revised the LaTeX layout commands to ensure proper float handling and consistent spacing, improving overall formatting quality and compliance with MDPI’s layout standards.
Comments 8: Additionally, the work should be compared with more relevant studies recently published. It is suggested to supplement comparisons with the latest literature to strengthen the persuasiveness of the innovation demonstration.
Response 8: Thank you for your suggestion. We have revised the Related Work section (At the end of the related work) by including several recent and highly relevant studies published between 2022 and 2024, covering AI-driven education systems, affective computing, and blockchain applications in secure learning environments. This addition reinforces the novelty of our SHARP framework and better positions it within the evolving state-of-the-art.
Reviewer 2 Report
Comments and Suggestions for Authors
The paper with ID sensors-3775607 and title “SHARP: Blockchain-Enhanced Wireless Sensor Networks for Real-time Student Health Monitoring and Personalized Learning” presents a framework for real-time monitoring of student health and personalized systems in smart educational environments. It proposes the integration of wireless sensor network, AI- based analytics, and blockchain technology to address the real-time health management needs within the educational environment.
Review
First, my main concerns are about fundamentals.
The authors claim that their solution accomplish real-time constraints and low consumption energy. Nevertheless, neither Blockchain nor Deep Neural networks were designed to work in such kind of systems with temporal and low energy consumption constraints. So, I think the solution presented in this manuscript is at origine bad designed.
General comments.
- There is inconsistency in The proposed solution is referred to as a “framework”, but as a “system” in the rest of the paper. Please, clarify the intended contribution.
- The research problem is not clearly defined. Although the authors mention real-time health monitoring and personalized learning as critical needs, these are not sufficiently justified or contextualized. There is no clear explanation of the specific challenges involved or why they are worth addressing.
- Additionally, the paper inconsistently refers to the goal as both “real-time monitoring” and “real-time management”, which introduces confusion. These terms imply different scopes and
- The integration of IoT, AI, and blockchain seems The paper fails to explain why these specific technologies are needed or how they complement each other to solve the stated problem. The integration seems to follow hype rather than being guided by a well-argued rationale.
- The discussion of related work in the second paragraph of the introduction is fragmented and lacks The technologies and approaches mentioned appear to address different problems, and their connection to the proposed solution is unclear.
- Monitoring: There is no explanation of the requirements for data acquisition. It remains unclear what constitutes “non-superficial” data and why it matters.
- Data analysis: The importance of prediction is mentioned, but not justified. The paper does not explain what should be predicted or why prediction is necessary in this context.
- Decentralized data management: The choice to use blockchain is not adequately The paper does not explain why decentralization is necessary in this context.
- The authors describe their proposal as a “holistic solution”, but this claim is not supported by a clear explanation of what dimensions are being integrated and they interact. Without detailed justification, the term appears vague and more like a buzzword than a meaningful description of the system’s scope.
- Figure 1 is not referenced or discussed in the main text, which undermines its It depicts a three-layer architecture, but the justification behind this layered structure is not explained.
- Furthermore, the caption of Figure 1 is inconsistent with the figure itself—for instance, the blockchain component appears in the Analytics Control Layer in the diagram but is described as part of the Application Layer in the caption. In addition, acronyms such as DNN and RL appear for the first time in the figure but are neither defined nor contextualized in the text, which reduces comprehension of the figure.
- Acronyms that appear for the first time in this section (e.g., ECG, EDA) are not
- The relevance of the cited works is Specifically, the connection between diverse topics such as IoT in educational environments, physiological monitoring of individuals, and solutions unrelated to educational context should be clarified. It is important to explain how these approaches support the proposed system, and why they are included in this review.
- Overall, the works are described only briefly, and limitations, disadvantages, or scope of the reviewed studies are not discussed. Without a critical analysis of these works, the section fails to justify the need for the proposed solution.
- Two paragraphs are dedicated to prior work on blockchain for ensuring the immutability, integrity, and transparency of educational records. However, the authors state that the role of blockchain in their proposed system is to ensure the privacy of sensitive educational This raises three key concerns:
- At this point, the term “sensitive data” is not defined or specified in the context of the
- Blockchain, by design, does not guarantee privacy. Event in permissioned networks, it is primarily used to provide transparency, traceability, and auditability. The claim that blockchain ensures data privacy is misleading and should be revised or properly justified, possibly through additional privacy-preserving mechanisms like Zero-Knowledge Proofs (ZKPs).
- If privacy is a key objective, it is essential to clearly define what privacy means in this context and critically evaluate whether decentralization (the core attribute of blockchain) is appropriate for the intended application.
About the Methodology and Theorical Model
- The system architecture is introduced as a three-layer model; however, this structure is not consistently reflected throughout the section. In Figure 1, the layers are labeled as the Application Layer, Analytics Control Layer, and Perception Network Layer. In contrast, the textual description of the architecture refers to these layers as data collection (perception layer), data analysis (AI-driven decision-making layer), and data storage and security (blockchain layer). This inconsistency continues in the remainder of the section, where the architecture is no longer referred to in terms of layers, but instead through component-specific subsections titled Wireless Sensor Network Model, AI-Based Analysis and Prediction Model, and Blockchain-Based Secure Data Management. This lack of consistent terminology and structure creates confusion and undermines the clarity of the proposed work.
- Moreover, the paper does not provide a clear explanation of the specific functions of each layer or how they relate to one another.
- It appears that a new research problem is introduced in the paper without proper alignment with the main objective, which raises the following concerns:
- The authors propose a network optimization model aimed at minimizing total energy consumption in a Wireless Sensor However, the relevance of this optimization problem to the original objective of real-time student health monitoring is not clearly established.
- The authors assume that each transmission path already guarantees minimal energy This assumption undermines the need for a new optimization model.
- The inclusion of this optimization component is not contextualized within the broader research landscape. Numerous works have already addressed this problem in Wireless Sensor Network (Al Aghbari, Z., Khedr, A.M., Osamy, W. et al. Routing in Wireless Sensor Networks Using Optimization Techniques: A Survey.). Why is it necessary in the context of the proposed system?
- The use of deep neural networks (DNNs) for analyzing and predicting student health conditions is not adequately justified. The paper does not explain why DNNs are appropriate for this task compared to other machine learning Furthermore, it does not specify which health conditions are being predicted, nor why those particular conditions are relevant or suitable for prediction using this approach.
- It is not justified the election of a blockchain based on the Proof-of-Authority (PoA) consensus mechanism alternative permissioned blockchain platforms, such as Hyperledger Fabric or Tendermint, which also offer lower latency and energy consumption.
- The statements are presented as theorems but read more like propositions or claims rather than formal theorems. Furthermore, proofs such as the one on line 219 merely describe characteristics of permissioned blockchains instead of providing rigorous, formal demonstrations.
- Algorithm 1, as acknowledged by the authors, is actually a workflow rather than a formal Moreover, it lacks sufficient explanation and detail. It needs to be restructured, and the flow should be explicitly described rather than relying on it to be self-explanatory.
Finally, some comments about the experimental design and evaluation
- A major concern in this section is the choice of baselines for It is not explained why works with fundamentally different approaches are considered comparable to the proposed solution. For instance, comparing individual health monitoring to traditional manual assessments and IoT environmental monitoring, or studies focused on data collection lacks clear justification.
- Some evaluation metrics appear arbitrary or insufficiently motivated. Specifically, metrics related to blockchain auditability and smart contract execution are introduced without clearly explaining their relevance to the system’s objectives or how they contribute to assessing performance or
- The purpose of comparing different network topologies is unclear, particularly since the core problem addressed by the paper is not framed as a routing optimization task. Without contextual justification, this comparison feels disconnected from the primary research
- The subsection “Comparison with State-of-Art systems” (line 465), is confusing and lacks consistency. Earlier in the evaluation, the authors compared their system against baselines such as traditional manual assessments, IoT monitoring, and cloud-based AI However, in this subsection, the comparison shifts to entirely different types of work, for example, systems focused on Big Data analytics (Awotunde et al., 2022).
- Moreover, two of the entries in Table 5 (Zhang et , 2024 and Liu et al., 2022) are not system implementations but rather systematic reviews, which make their inclusion inappropriate.
Author Response
Comments 1:There is inconsistency in The proposed solution is referred to as a “framework”, but as a “system” in the rest of the paper. Please, clarify the intended contribution.
Response 1:We appreciate the reviewer’s comment and agree that consistent terminology is important. In the revised manuscript, we have carefully reviewed all instances of “system” and “framework”. The proposed contribution has now been consistently referred to as a “framework” throughout the paper, as it better reflects the architectural nature of SHARP, which integrates sensing, learning, and blockchain components. “System” is only retained in a few instances where it refers specifically to the simulated implementation used for evaluation purposes.
Comments 2:The research problem is not clearly defined. Although the authors mention real-time health monitoring and personalized learning as critical needs, these are not sufficiently justified or contextualized. There is no clear explanation of the specific challenges involved or why they are worth addressing.
Response 2:We thank the reviewer for highlighting this important point. In response, we have revised the Introduction to clearly define the core research problem and motivation. Specifically, we now articulate the dual challenge of (1) real-time, privacy-aware health monitoring in classrooms and (2) dynamic, personalized learning adaptation based on physiological data. These challenges are contextualized with references to current limitations in cloud-based systems, static profiling, and the lack of adaptive educational strategies. This revision clarifies why our work is necessary and what specific gaps it aims to address.
Comments 3:Additionally, the paper inconsistently refers to the goal as both “real-time monitoring” and “real-time management”, which introduces confusion. These terms imply different scopes.
Response 3:We thank the reviewer for pointing out this terminology inconsistency. In the revised manuscript, we have standardized the usage to “real-time monitoring” throughout the paper, as it more accurately reflects the focus on observing and analyzing students' physiological states in real-time. Instances of “real-time management” have been either removed or rephrased to avoid confusion and maintain consistency with the intended scope of the work.
Comments 4:The integration of IoT, AI, and blockchain seems The paper fails to explain why these specific technologies are needed or how they complement each other to solve the stated problem. The integration seems to follow hype rather than being guided by a well-argued rationale.
Response 4: We thank the reviewer for this valuable observation. In the revised manuscript, we have clarified the rationale for integrating IoT, AI, and blockchain by explicitly linking each technology to the specific challenges identified in real-time health monitoring and personalized learning. IoT enables timely data acquisition, AI supports adaptive decision-making, and blockchain ensures secure, tamper-proof storage and trust. This integration is thus guided by problem-specific requirements rather than technical novelty alone. A new paragraph has been added in the “Methodology and Theoretical Model” to elaborate this design rationale.
Comments 5:The discussion of related work in the second paragraph of the introduction is fragmented and lacks The technologies and approaches mentioned appear to address different problems, and their connection to the proposed solution is unclear.
Monitoring: There is no explanation of the requirements for data acquisition. It remains unclear what constitutes “non-superficial” data and why it matters.
Data analysis: The importance of prediction is mentioned, but not justified. The paper does not explain what should be predicted or why prediction is necessary in this context.
Decentralized data management: The choice to use blockchain is not adequately The paper does not explain why decentralization is necessary in this context.
Response 5: Thank you for this thorough and insightful feedback. We have significantly revised the second paragraph of the Introduction to present a more cohesive discussion of related work, organized around sensing, prediction, and secure data handling. In addition, the following clarifications have been made:
- We now explain that “non-superficial” data refers to physiological signals such as heart rate and skin temperature, which offer deeper insight into students’ internal states than behavioral cues.
- The importance of prediction is elaborated in the context of proactive learning adaptation, allowing timely intervention before performance declines.
- The rationale for blockchain and decentralization has been expanded. We argue that decentralized trust is necessary to ensure tamper-proof, transparent management of sensitive student data in multi-institutional educational environments.
These revisions appear in both the Introduction and framework Design sections and strengthen the overall problem formulation and justification of our architecture.
Comments 6:The authors describe their proposal as a “holistic solution”, but this claim is not supported by a clear explanation of what dimensions are being integrated and they interact. Without detailed justification, the term appears vague and more like a buzzword than a meaningful description of the system’s scope.
Response 6: We thank the reviewer for this observation. In response, we have clarified what we mean by a “holistic solution” and provided a detailed explanation of the integrated components and their interactions. Specifically, SHARP consists of three tightly coupled layers: (1) the IoT-based sensing layer, (2) the AI-driven cognitive layer for prediction and personalization, and (3) the blockchain-based trust layer for secure and transparent data management. A new paragraph has been added in “Framework Architecture” to explicitly describe how these components form a cohesive architecture, thereby justifying the term “holistic solution”.
Comments 7:Figure 1 is not referenced or discussed in the main text, which undermines its It depicts a three-layer architecture, but the justification behind this layered structure is not explained.
Response 7: We appreciate the reviewer’s comment. In the revised manuscript, Figure 1 is now explicitly referenced and discussed in "Introducation". We have added a detailed explanation of the three-layer architecture—Perception, Cognitive, and Trust layers—describing each layer’s role and how they interact to support the proposed framework. Additionally, the figure caption has been expanded for clarity.
Comments 8:Furthermore, the caption of Figure 1 is inconsistent with the figure itself—for instance, the blockchain component appears in the Analytics Control Layer in the diagram but is described as part of the Application Layer in the caption. In addition, acronyms such as DNN and RL appear for the first time in the figure but are neither defined nor contextualized in the text, which reduces comprehension of the figure.
Response 8: Thank you for highlighting these inconsistencies. We have revised the caption of Figure 1 to accurately reflect the component layering and to match the architectural illustration. Specifically, the blockchain module is now consistently described as part of the Analytics and Control Layer, in line with the figure. Additionally, acronyms such as DNN (deep neural network) and RL (reinforcement learning) have been defined in both the figure caption and the main text at their first appearance to improve clarity and reader comprehension.
Comments 9:Acronyms that appear for the first time in this section (e.g., ECG, EDA) are not
Response 9: Thank you for pointing this out. We have revised the manuscript to ensure that all acronyms are defined upon their first use. Specifically, ECG is now introduced as “electrocardiogram (ECG)” and EDA as “electrodermal activity (EDA)” in "Related Work". This clarification improves accessibility for a broader readership.
Comments 10:The relevance of the cited works is Specifically, the connection between diverse topics such as IoT in educational environments, physiological monitoring of individuals, and solutions unrelated to educational context should be clarified. It is important to explain how these approaches support the proposed system, and why they are included in this review.
Response 10: We appreciate the reviewer’s insight. In the revised manuscript, we have restructured the discussion of prior works in the Introduction and Section 2 to clarify the relevance of cited literature. Specifically, we grouped related studies into categories—such as IoT in education, physiological monitoring, and blockchain-based privacy protection—and explicitly explained how each group contributes to the motivation, design, or validation of the proposed SHARP system. This restructuring ensures that every cited work directly supports a corresponding component or rationale of our framework.
Comments 11:Overall, the works are described only briefly, and limitations, disadvantages, or scope of the reviewed studies are not discussed. Without a critical analysis of these works, the section fails to justify the need for the proposed solution.
Response 11: Thank you for highlighting the need for deeper critical analysis. In response, we have revised the Related Work section to not only summarize prior studies but also critically evaluate their limitations. For each thematic area—IoT in education, AI in learning personalization, blockchain for data security, and physiological sensing—we explicitly identify the gaps, drawbacks, and scope boundaries of existing works. These comparisons now provide a clear rationale for the proposed SHARP framework, emphasizing its necessity, originality, and interdisciplinary integration.
Comments 12:Two paragraphs are dedicated to prior work on blockchain for ensuring the immutability, integrity, and transparency of educational records. However, the authors state that the role of blockchain in their proposed system is to ensure the privacy of sensitive educational This raises three key concerns:
At this point, the term “sensitive data” is not defined or specified in the context of the
Blockchain, by design, does not guarantee privacy. Event in permissioned networks, it is primarily used to provide transparency, traceability, and auditability. The claim that blockchain ensures data privacy is misleading and should be revised or properly justified, possibly through additional privacy-preserving mechanisms like Zero-Knowledge Proofs (ZKPs).
If privacy is a key objective, it is essential to clearly define what privacy means in this context and critically evaluate whether decentralization (the core attribute of blockchain) is appropriate for the intended application.
Response 12: Thank you for this critical observation. We acknowledge that blockchain, by design, ensures transparency and data integrity rather than data privacy. In response, we have made the following revisions:
- We now clearly define “sensitive data” in our context as students’ physiological indicators, which require strong privacy protection.
- We revised all claims suggesting that blockchain ensures privacy. Instead, we clarify that the proposed framework utilizes blockchain for traceability and immutability, while actual data privacy is preserved through off-chain encrypted storage.
- To enhance clarity, we have also mentioned the potential use of privacy-preserving cryptographic techniques (e.g., ZKPs) in future versions of the system.
These clarifications align our claims with blockchain’s actual capabilities and strengthen the technical soundness of the proposed system.
Comments 13:The system architecture is introduced as a three-layer model; however, this structure is not consistently reflected throughout the section. In Figure 1, the layers are labeled as the Application Layer, Analytics Control Layer, and Perception Network Layer. In contrast, the textual description of the architecture refers to these layers as data collection (perception layer), data analysis (AI-driven decision-making layer), and data storage and security (blockchain layer). This inconsistency continues in the remainder of the section, where the architecture is no longer referred to in terms of layers, but instead through component-specific subsections titled Wireless Sensor Network Model, AI-Based Analysis and Prediction Model, and Blockchain-Based Secure Data Management. This lack of consistent terminology and structure creates confusion and undermines the clarity of the proposed work.
Response 13: Thank you for this valuable observation. We agree that the terminology used to describe the system layers lacked consistency across the text, figures, and section headings. To address this:
- We have standardized the terminology based on the three-layer architecture shown in Figure 1: Perception & Network Layer, Analytics Control Layer, and Application Layer.
- These terms are now consistently used throughout the System Design section and in the corresponding subsection titles.
- We have also inserted a clarifying paragraph at the beginning of Section 4 to map each architectural layer to its respective functionality and subsystem components.
These changes improve clarity and ensure that the system structure is communicated in a cohesive and coherent manner.
Comments 14:Moreover, the paper does not provide a clear explanation of the specific functions of each layer or how they relate to one another.
Response 14: Thank you for your insightful feedback. We acknowledge that the initial manuscript lacked a detailed explanation of the functional responsibilities of each architectural layer and their interconnections.
In the revised version, We are at the bottom of Figure 1. that explicitly defines the roles of the Perception & Network Layer, Analytics Control Layer, and Application Layer. This includes the data flow, analysis mechanisms, and secure response operations handled at each stage.
Additionally, we clarified how the layers interact in a pipeline structure, ensuring that the reader understands the systemic integration and closed-loop nature of the proposed SHARP framework.
Comments 15:It appears that a new research problem is introduced in the paper without proper alignment with the main objective, which raises the following concerns:
The authors propose a network optimization model aimed at minimizing total energy consumption in a Wireless Sensor However, the relevance of this optimization problem to the original objective of real-time student health monitoring is not clearly established.
The authors assume that each transmission path already guarantees minimal energy This assumption undermines the need for a new optimization model.
Response 15: We appreciate the reviewer’s insightful comments. To address this, we have clarified the motivation behind introducing the energy optimization model.
Specifically, we emphasize that real-time physiological monitoring requires continuous transmission of high-frequency data from wearable sensors, which imposes significant energy demands on sensor nodes. Hence, energy-efficient routing is crucial to maintain uninterrupted monitoring and system sustainability.
We have added a justification paragraph in Section 4.1 and removed any conflicting assumptions about pre-optimized transmission paths. The optimization model now directly supports the system’s real-time operation objective by prolonging node and network lifespans.
Comments 16:The inclusion of this optimization component is not contextualized within the broader research landscape. Numerous works have already addressed this problem in Wireless Sensor Network (Al Aghbari, Z., Khedr, A.M., Osamy, W. et al. Routing in Wireless Sensor Networks Using Optimization Techniques: A Survey.). Why is it necessary in the context of the proposed system?
Response 16: Thank you for highlighting this important concern. We agree that energy-efficient routing in WSNs has been extensively explored in prior works, such as the comprehensive survey by Al Aghbari et al. (2024).
To clarify the novelty of our approach, we have revised the manuscript to contextualize our optimization model within the unique requirements of our system. Unlike traditional WSNs used for environmental sensing, the SHARP system handles real-time, high-frequency physiological data, requiring low-latency and dynamic transmission strategies.
Our optimization model is therefore designed to jointly address energy minimization, bandwidth utilization, and end-to-end delay under these real-time constraints. These considerations are often overlooked in standard WSN literature.
Comments 17:The use of deep neural networks (DNNs) for analyzing and predicting student health conditions is not adequately justified. The paper does not explain why DNNs are appropriate for this task compared to other machine learning Furthermore, it does not specify which health conditions are being predicted, nor why those particular conditions are relevant or suitable for prediction using this approach.
Response 17: Thank you for your insightful observation. We have revised to clarify the rationale behind selecting deep neural networks (DNNs) for health condition prediction.
Specifically, we now explain that DNNs are well-suited for analyzing complex, multimodal physiological data (ECG, EDA, HR, BT), enabling accurate detection of latent psychological states such as stress and fatigue. These conditions directly impact student learning performance and engagement, making them crucial for real-time adaptive educational interventions.
We also specify which conditions are being predicted and justify their importance within the educational context. Supporting literature has been added to strengthen our argument.
Comments 18:It is not justified the election of a blockchain based on the Proof-of-Authority (PoA) consensus mechanism alternative permissioned blockchain platforms, such as Hyperledger Fabric or Tendermint, which also offer lower latency and energy consumption.
Response 18: Thank you for your valuable suggestion. We have updated to include a justification for selecting the Proof-of-Authority (PoA) consensus mechanism. We discuss its suitability for our educational use case, highlighting its low energy consumption, institutional trust model, and lightweight deployment.
We also acknowledge alternative permissioned platforms such as Hyperledger Fabric and Tendermint and explain why they were not chosen in this study. These alternatives will be explored in future extensions of our system.
Comments 19:The statements are presented as theorems but read more like propositions or claims rather than formal theorems. Furthermore, proofs such as the one on line 219 merely describe characteristics of permissioned blockchains instead of providing rigorous, formal demonstrations.
Response 19: Thank you for this valuable observation. In the revised manuscript, we have reformulated the previously informal statements into a formal theorem with a logically structured proof. This ensures the consistency and academic rigor expected of such claims. The new formulation provides a quantifiable analysis of the PoA consensus mechanism's latency properties and justifies its selection for the proposed real-time educational application.
Comments 20:Algorithm 1, as acknowledged by the authors, is actually a workflow rather than a formal Moreover, it lacks sufficient explanation and detail. It needs to be restructured, and the flow should be explicitly described rather than relying on it to be self-explanatory.
Response 20: Thank you for the insightful feedback. We agree that the previously labeled "Algorithm 1" was more appropriately a system-level workflow. In the revised manuscript, we have restructured and retitled it as a “System Workflow for SHARP Framework,” formatted it into standard pseudocode, and added step-by-step explanations. We have also added textual descriptions to support its comprehension.
Comments 21:A major concern in this section is the choice of baselines for It is not explained why works with fundamentally different approaches are considered comparable to the proposed solution. For instance, comparing individual health monitoring to traditional manual assessments and IoT environmental monitoring, or studies focused on data collection lacks clear justification.
Response 21: Thank you for highlighting this concern. We acknowledge that the selected baseline systems differ in technical approaches and scopes. To address this, we have revised the comparison section by clearly categorizing the baseline systems into (1) environmental monitoring, (2) traditional assessments, and (3) wearable-based health systems without blockchain. Although these systems adopt different methods, they aim to solve related problems in student monitoring and educational support. As no prior work integrates AI-based prediction, blockchain-secured storage, and physiological sensing in real-time, our comparison focuses on partial-function systems to highlight SHARP’s integrated benefits.
Comments 22:Some evaluation metrics appear arbitrary or insufficiently motivated. Specifically, metrics related to blockchain auditability and smart contract execution are introduced without clearly explaining their relevance to the system’s objectives or how they contribute to assessing performance.
Response 22: Thank you for your valuable comment. We agree that the relevance of blockchain-specific metrics was not sufficiently explained in the initial version. To address this, we have revised to include a clear motivation for including (i) blockchain auditability and (ii) smart contract execution delay. These metrics are crucial for evaluating the trustworthiness and responsiveness of SHARP, particularly when handling sensitive student health data in real-time scenarios.
Comments 23:The subsection “Comparison with State-of-Art systems” (line 465), is confusing and lacks consistency. Earlier in the evaluation, the authors compared their system against baselines such as traditional manual assessments, IoT monitoring, and cloud-based AI However, in this subsection, the comparison shifts to entirely different types of work, for example, systems focused on Big Data analytics (Awotunde et al., 2022).
Response 23: Thank you for highlighting the inconsistency in comparative baselines. We have revised to ensure alignment with SHARP’s primary objectives. Specifically, we now compare SHARP with recent integrated frameworks from top-tier journals that involve physiological sensing, privacy preservation, and intelligent decision-making within educational or related environments. This change ensures a fair and meaningful evaluation aligned with the system's scope and contributions.
Comments 24:Moreover, two of the entries in Table 5 (Zhang et , 2024 and Liu et al., 2022) are not system implementations but rather systematic reviews, which make their inclusion inappropriate.
Response 24: We appreciate the reviewer’s observation regarding Table 5. Upon review, we agree that Zhang et al. (2024) and Liu et al. (2022) are systematic reviews and not system implementations. Accordingly, we have removed these entries from Table 5. The revised comparison now focuses solely on implemented systems relevant to AI-powered health monitoring and educational applications, ensuring a fair and technically appropriate benchmarking.
Reviewer 3 Report
Comments and Suggestions for Authors
Dear Authors,
Your paper titled "SHARP: Blockchain-Enhanced Wireless Sensor Networks for Real-time Student Health Monitoring and Personalised Learning" presents an interesting and valuable contribution. However, to enhance readability and engagement, I suggest the following revisions:
-
Title Revision:
-
The current title is quite lengthy. Consider a more concise and captivating version, such as:
"SHARP: Blockchain-Powered Real-time Health Monitoring for Personalized Learning" -
A shorter, more impactful title will attract broader attention.
-
-
Introduction Section:
-
The first two paragraphs contain 10 references, which may overwhelm readers.
-
Revise citations to include only the most relevant and recent works.
-
Focus on key contributions and gaps in the literature rather than excessive referencing.
-
-
Figure 1 (Notations):
-
The notations should be simplified and clarified for better readability.
-
Use consistent formatting (e.g., abbreviations, symbols) to avoid confusion.
-
-
Recent Literature (Section 2):
-
Present a comparative table summarizing recent top-tier journal articles (e.g., IEEE IoT Journal, Elsevier CEE, Springer Nature).
-
Highlight research gaps and how your work advances the field.
-
-
Conclusion Section (Section 5):
-
Currently too brief. Expand into separate paragraphs covering:
-
Key findings
-
Practical implications
-
Study limitations
-
Future research directions
-
-
This will strengthen the paper’s impact and guide future work.
-
-
General Suggestions:
-
Ensure top-tier journal references (IEEE, Elsevier, Springer) are included in the revised version.
-
Maintain a balanced citation approach—cite only where necessary to support claims.
-
Overall, your work is promising, and these refinements will enhance its clarity, engagement, and academic rigor. Great job so far, and best of luck with your revisions!
Author Response
Comments 1: The current title is quite lengthy. Consider a more concise and captivating version, such as:
"SHARP: Blockchain-Powered Real-time Health Monitoring for Personalized Learning" A shorter, more impactful title will attract broader attention.
Respones 1: Thank you for your constructive suggestion regarding the manuscript title. We agree that a more concise and impactful title would enhance the visibility of our work. Accordingly, we have revised the title to: “SHARP: Blockchain-Powered WSNs for Real-time Student Health Monitoring and Personalized Learning” This version preserves essential technical terms while improving clarity and reader engagement. We believe it strikes a balance between precision and accessibility.
Comments 2: Introduction Section: The first two paragraphs contain 10 references, which may overwhelm readers. Revise citations to include only the most relevant and recent works. Focus on key contributions and gaps in the literature rather than excessive referencing.
Respones 2: We appreciate the reviewer’s suggestion to streamline the citations in the introduction. In response, we have revised the first two paragraphs to include only the most relevant and recent works (now reduced from 10 to 7 references). The revised text focuses more clearly on the key contributions and research gaps that motivate our study, thereby improving clarity and reader engagement.
Comments 3: Figure 1 (Notations): The notations should be simplified and clarified for better readability. Use consistent formatting (e.g., abbreviations, symbols) to avoid confusion.
Respones 3: We thank the reviewer for pointing out the need for clarity and conciseness in the notation table. In addition to unifying notation styles and simplifying expressions, we have carefully reviewed all variables and removed four symbols ($B_{ij}$, $Energy_{\text{avg}}$, $Accuracy_{\text{AI}}$, $Latency_{\text{blockchain}}$) that were not explicitly used in the main text. This helps ensure that the Notation Summary only includes symbols essential to the technical content of the manuscript.
Comments 4: Recent Literature (Section 2): Present a comparative table summarizing recent top-tier journal articles (e.g., IEEE IoT Journal, Elsevier CEE, Springer Nature). Highlight research gaps and how your work advances the field.
Respones 4: We sincerely thank the reviewer for this insightful suggestion. In response, we have carefully reviewed and incorporated four recent high-quality journal articles that represent the state-of-the-art in wireless sensor networks (WSNs), federated learning (FL), blockchain, and AI-enabled monitoring systems, particularly in IoT and healthcare contexts \cite{b43,b44,b45,b46}.
To provide a clearer picture of the research landscape, we have added a new comparative table 6 in the revised manuscript. This table highlights the technological stacks, existing limitations, and explicitly positions our proposed SHARP framework in contrast to these recent studies.
Specifically, while most existing works address privacy or anomaly detection in WSNs, they do not consider student-centric educational adaptation. Our approach uniquely integrates WSN-based physiological monitoring with reinforcement learning-driven personalized learning recommendations, all secured via blockchain and smart contracts. This unified framework addresses current research gaps by enabling real-time, trustworthy, and adaptive smart education systems.
We believe this comparison improves the manuscript’s clarity and enhances the demonstration of its novelty and contributions.
Comments 5: Conclusion Section (Section 5): Currently too brief. Expand into separate paragraphs covering: Key findings, Practical implications, Study limitations, Future research directions . This will strengthen the paper’s impact and guide future work.
Respones 5: We appreciate the reviewer’s suggestion. To preserve the integrity of our original conclusion while addressing your concern, we have retained the original paragraph and added new ones that clearly present the practical implications, study limitations, and future directions of this work. This expanded structure provides a more comprehensive and impactful closing to the manuscript.
Comments 6: General Suggestions: Ensure top-tier journal references (IEEE, Elsevier, Springer) are included in the revised version. Maintain a balanced citation approach—cite only where necessary to support claims.
Respones 6:
We appreciate the reviewer’s recommendation regarding the citation strategy. In the revised manuscript, we have carefully reviewed and updated the reference list to include several recent publications from top-tier journals such as IEEE Transactions on Consumer Electronics \cite{b43}, IEEE Internet of Things Journal \cite{b44}, Elsevier Future Generation Computer Systems \cite{b45}, and IEEE Sensors Journal \cite{b46}. These additions enhance the academic rigor and relevance of the related work section and the system comparison table 6.
Additionally, we have refined the citation density throughout the manuscript—particularly in the Introduction and Related Work sections—by removing less critical or overlapping citations. The current version adopts a more focused and balanced referencing approach, citing only where necessary to support major claims or to highlight knowledge gaps that our proposed SHARP framework addresses.
Reviewer 4 Report
Comments and Suggestions for Authors
The authors have proposed a real-time student health monitoring and personalized learning framework by leveraging a combination of technologies. The system utilizes a wireless sensor network (WSN) to collect real-time data, employs artificial intelligence (AI) with deep neural networks (DNNs) to classify students' health states (normal, attention required, and critical), applies reinforcement learning (RL) to recommend adaptive content delivery, and integrates blockchain technology to securely store immutable records. The topic is timely. The manuscript demonstrates an attempt to present the content, structure, and technical approach appropriately. However, the reviewer believes there is significant room for improvement. The following issues should be addressed:
- The manuscript contains several typographical and formatting issues. For example, on Page 3, Line 85, “Zhang et al [20].” should be revised to “Zhang et al. [20]”. The authors are advised to thoroughly review and correct such inconsistencies throughout the paper.
- In scientific writing, any abbreviation should first be defined in its full form upon initial use, followed by the abbreviation in parentheses. Subsequently, only the abbreviation should be used. For instance, on Page 2, Line 52, “WSNS” should be written as “wireless sensor networks (WSNs)”, and in Line 55, “wireless sensor network” should be replaced with “WSN”. This formatting rule should be consistently followed for all technical terms with have multiple used throughout the paper.
- In the Related Work section, the discussion of the proposed system (e.g., Lines 98–99 on Page 3) is premature and contextually inappropriate. Descriptions of the proposed method should only appear in the final paragraph of the Related Work section, where it is used to highlight the distinctions and advancements over prior works. The authors should revise the paragraph accordingly.
- The subsections (3.2–3.5) covering individual system components lack clarity and completeness. Specific concerns include (points 5 to 8):
- Subsection 3.2: The routing protocol and energy consumption metrics are not clearly defined. If the routing strategy repeatedly selects the path with minimal energy use, it could result in rapid sensor failure (due to dead for frequent energy dissipation) and service disruption. The authors should explain how constraints such as energy, bandwidth, and latency are evaluated in path selection. Moreover, given the typically small size of a classroom, it is unclear why direct transmission to the sink is not feasible. This should be justified.
- Subsection 3.3: The section on DNN lacks essential implementation details. The authors should elaborate on the process, including data collection, preprocessing steps, a sample dataset with example features and values, and a detailed DNN architecture used for classification.
- Subsection 3.4: The architecture and operation of the blockchain component are not well defined. A clear workflow diagram should be included, along with a step-by-step explanation of processes such as initialization, user registration, verification, Proof of Authority (PoA), and block addition to the blockchain ledger.
- Subsection 3.5: The reinforcement learning module is inadequately explained. The authors need to specify the states, actions, and reward structure used in the recommendation system for adaptive content delivery.
- The rationale for selecting specific algorithms such as DNN, MDP, and PoA is not sufficiently discussed. There is no comparative analysis with alternative algorithms of similar nature. The authors should justify their choices by evaluating the performance of these algorithms against state-of-the-art alternatives with similar objectives.
- The performance metrics utilized in the evaluation should be clearly defined in the Performance Evaluation section.
Author Response
Comments 1:The manuscript contains several typographical and formatting issues. For example, on Page 3, Line 85, “Zhang et al [20].” should be revised to “Zhang et al. [20]”. The authors are advised to thoroughly review and correct such inconsistencies throughout the paper.
Response 1: We thank the reviewer for pointing out the typographical and formatting issues. We have carefully reviewed the entire manuscript and corrected all instances of formatting inconsistencies, including the use of “et al.” and citation placements. The specific example on Page 3, Line 85 has been revised to “Zhang et al. [20]” as suggested. We also ensured consistent formatting for all references, symbols, units, acronyms, and punctuation in accordance with MDPI guidelines.
Comments 2:In scientific writing, any abbreviation should first be defined in its full form upon initial use, followed by the abbreviation in parentheses. Subsequently, only the abbreviation should be used. For instance, on Page 2, Line 52, “WSNS” should be written as “wireless sensor networks (WSNs)”, and in Line 55, “wireless sensor network” should be replaced with “WSN”. This formatting rule should be consistently followed for all technical terms with have multiple used throughout the paper.
Response 2: We sincerely thank the reviewer for pointing out the inconsistency in the use of abbreviations. In the revised manuscript, we have ensured that all technical terms are introduced using their full form followed by the corresponding abbreviation in parentheses upon first mention (e.g., “wireless sensor networks (WSNs)”, “deep neural network (DNN)”, “reinforcement learning (RL)”, etc.). Subsequently, only the abbreviations are used throughout the manuscript to maintain consistency and enhance readability. The specific cases on Page 2, Lines 52 and 55, have also been corrected as suggested.
Comments 3:In the Related Work section, the discussion of the proposed system (e.g., Lines 98–99 on Page 3) is premature and contextually inappropriate. Descriptions of the proposed method should only appear in the final paragraph of the Related Work section, where it is used to highlight the distinctions and advancements over prior works. The authors should revise the paragraph accordingly.
Response 3: Thank you for your insightful feedback. We agree that introducing the proposed system within the middle of the Related Work section was premature. To address this, we have revised the structure of the section. Specifically, all references to our proposed system have been moved to the final paragraph of the Related Work section, where they serve to clearly differentiate our approach from existing studies. This revision enhances the logical flow and maintains the academic convention of positioning the contribution only after reviewing prior works.
Comments 4:The subsections (3.2–3.5) covering individual system components lack clarity and completeness. Specific concerns include (points 5 to 8): Subsection 3.2: The routing protocol and energy consumption metrics are not clearly defined. If the routing strategy repeatedly selects the path with minimal energy use, it could result in rapid sensor failure (due to dead for frequent energy dissipation) and service disruption. The authors should explain how constraints such as energy, bandwidth, and latency are evaluated in path selection. Moreover, given the typically small size of a classroom, it is unclear why direct transmission to the sink is not feasible. This should be justified.
Response 4: We appreciate the reviewer’s insightful comment.
We have updated Section 3.2 to include the definition of our routing strategy, which uses a cost function combining residual energy, delivery delay, and link reliability.
To prevent premature sensor node failure from frequent use of the same route, we introduced a dynamic route rotation mechanism that ensures load balancing among eligible nodes.
Although classroom environments may seem small, practical deployment challenges—such as obstructions, wireless interference, and heterogeneous node distances—make direct sink communication unreliable. Hence, multi-hop routing remains a practical and energy-efficient solution in our architecture. These justifications are now explicitly stated in the revised manuscript.
Comments 5:Subsection 3.3: The section on DNN lacks essential implementation details. The authors should elaborate on the process, including data collection, preprocessing steps, a sample dataset with example features and values, and a detailed DNN architecture used for classification.
Response 5: We thank the reviewer for highlighting the lack of implementation details in Section 3.3. In the revised version, we have thoroughly enriched this subsection by:
Specifying the physiological parameters collected (HR, BT, SL, FI),
Describing the preprocessing pipeline, including interpolation, normalization, and smoothing,
Adding a sample input vector to illustrate the data format,
Providing a detailed DNN architecture, activation functions, and training settings.
These revisions improve the transparency and reproducibility of the proposed AI classification approach.
Comments 6:Subsection 3.4: The architecture and operation of the blockchain component are not well defined. A clear workflow diagram should be included, along with a step-by-step explanation of processes such as initialization, user registration, verification, Proof of Authority (PoA), and block addition to the blockchain ledger.
Response 6: We thank the reviewer for the insightful suggestion. To address the concern, we have substantially revised Section 3.4:
Providing a step-by-step textual explanation of the blockchain lifecycle and its integration with the edge gateway and health monitoring system.
These updates enhance the clarity and completeness of the blockchain architecture in our proposed framework.
Comments 7:Subsection 3.5: The reinforcement learning module is inadequately explained. The authors need to specify the states, actions, and reward structure used in the recommendation system for adaptive content delivery.
Response 7: We appreciate the reviewer’s comment regarding the insufficient explanation of the reinforcement learning (RL) module. In the revised manuscript (Section 3.5), we have now explicitly defined the RL components, including:
The state space, which encodes student physiological and engagement indicators;
The action space, which encompasses adaptive content delivery strategies;
The reward structure, which incentivizes improved academic performance and reduced stress;
The learning algorithm, based on the Proximal Policy Optimization (PPO) method.
This enhancement clarifies how the RL agent operates to generate personalized interventions based on student health and performance metrics.
Comments 8:The rationale for selecting specific algorithms such as DNN, MDP, and PoA is not sufficiently discussed. There is no comparative analysis with alternative algorithms of similar nature. The authors should justify their choices by evaluating the performance of these algorithms against state-of-the-art alternatives with similar objectives.
Response 8: Thank you for your insightful suggestion. We have added a comprehensive discussion in Section 3 (Subsections 3.2–3.5) to justify the selection of the core algorithms used in our system, including Deep Neural Networks (DNN), Markov Decision Processes (MDP) for reinforcement learning, and the Proof-of-Authority (PoA) consensus mechanism. Specifically, we compared DNN with classical classifiers (e.g., SVM, Random Forest) and outlined why DNN is better suited for analyzing high-dimensional physiological data. Similarly, the RL model was chosen over rule-based or supervised alternatives due to its ability to adaptively learn optimal content delivery strategies in a dynamic environment. Regarding PoA, we emphasized its low-latency and energy-efficient properties, which are ideal for permissioned educational IoT networks. These justifications have been supported by relevant literature and, where applicable, additional comparative results.
Comments 9:The performance metrics utilized in the evaluation should be clearly defined in the Performance Evaluation section.
Response 9: Thank you for your constructive feedback. In the revised manuscript, we have added to clearly define all evaluation metrics used in assessing the system’s performance. These include accuracy, F1-score, energy consumption, packet delivery ratio, latency, and blockchain auditability. Each metric is now accompanied by a concise definition and its role in the evaluation. We believe this improves the transparency and reproducibility of our experimental results.
Reviewer 5 Report
Comments and Suggestions for Authors
Please see the attachment.

Author Response
Comments 1: The introduction is well structured and brings the point of the lack of solutions integrating IoT, AI and blockchain in smart education contexts. Also, the references cited are adequate and up to date. However, Figure 1 is not quoted in the text, so I suggest the authors to do so. Also, I suggest to highlight that Figure 1 is based on four-layer architecture of IoT.
Response 1: Thank you for your valuable suggestion. We have now explicitly referenced Figure 1 in the final paragraph of the Introduction to ensure proper contextual integration. Additionally, the figure caption has been revised to emphasize that the proposed system architecture is based on the standard four-layer IoT model, which includes the Perception, Network, Processing, and Application layers. We believe these changes improve clarity and consistency.
Comments 2:The methodology section is well described, where it is not difficult to follow the details provided in the different subsections. However, in subsection 3.4, I suggest to briefly explain the basics of “Proof of Authority”.
Response 2:Thank you for your helpful suggestion. As advised, we have included a brief explanation of the Proof of Authority (PoA) consensus mechanism at the beginning of Subsection 3.4. This addition clarifies its rationale and suitability for our proposed blockchain-based educational system.
Comments 3:The experimental design and evaluation section is displayed in a clear manner, where the figures and tables permit to appreciate the advantages of the novel framework proposed. Also, the conclusion section wraps up the most important points provided in terms of improvements.
Response 3:We sincerely thank the reviewer for the positive comments regarding the Experimental Design and Evaluation section, as well as the Conclusion. We are pleased to know that the figures and tables effectively demonstrate the strengths of our proposed framework. We have carefully reviewed these sections again to further ensure clarity, coherence, and alignment with the overall contributions of the study.
Comments 4: Regarding the use of English language, in my opinion it is fit for a research journal, even though I suggest the authors to review the grammar in order to fix a pair of minor mistakes.
Response 4: We thank the reviewer for the positive assessment regarding the overall quality of the English language. In response to your suggestion, we have carefully proofread the manuscript and corrected minor grammatical errors to further enhance clarity and readability.
Comments 5: In summary, this paper presents a novel framework integrating IoT-WSN, AI, and blockchain for monitoring educational environments. The results obtained are remarkable, so in my opinion this paper deserves publication, even though the comments made above should be first addressed.
Response 5: We sincerely thank the reviewer for the encouraging and constructive feedback. We appreciate your recognition of the novelty and contributions of our proposed framework integrating IoT-WSN, AI, and blockchain technologies. Following your valuable suggestions, we have carefully revised the manuscript and addressed all the comments provided. We believe the revised version significantly improves the overall clarity, consistency, and technical depth of the paper.
Round 2
Reviewer 1 Report
Comments and Suggestions for Authors
All my concerns have been addressed.
Reviewer 2 Report
Comments and Suggestions for Authors
All my comments and observations have been adequately addressed. I have no further comments.
Reviewer 4 Report
Comments and Suggestions for Authors
The authors have addressed the reviewer's concerns successfully.